# Robust In-Context Reinforcement Learning Under Reward Poisoning Attacks

**Paulius Sasnauskas** [1 2 †]  **Yiğit Yalın** [3]  **Goran Radanović** [3]

## Abstract

We study the corruption-robustness of in-context reinforcement learning (ICRL), focusing on the Decision-Pretrained Transformer (DPT, Lee et al., 2023). To address the challenge of reward poisoning attacks targeting the DPT, we propose a novel adversarial training framework, called Adversarially Trained DPT (AT-DPT). Our method simultaneously trains a population of attackers to minimize the true reward of the DPT by poisoning environment rewards, and a DPT model to infer optimal actions from the poisoned data. We evaluate the effectiveness of our approach against standard bandit algorithms, including robust baselines designed to handle reward contamination. Our results show that AT-DPT significantly outperforms them in bandit settings under a learned attacker, and generalizes to more complex environments such as adaptive attackers and MDPs. It shows promise in ICRL as a meta-RL approach to learning effective corruption-robust algorithms.

## 1. Introduction

Recent years have shown the impressive capabilities of transformer-based models on a range of tasks (Vaswani et al., 2017; Raffel et al., 2020). The community has been shifting from single-task learning, to multi-task learning, and even multi-domain learning (Reed et al., 2022). This has been made possible in part due to in-context learning, also called few-shot learning (Brown et al., 2020), which allows models to adapt to new tasks simply by reading a handful of examples in the prompt, rather than requiring parameter updates. Recently, transformers and in-context learning have found growing use in decision-making tasks, particularly in reinforcement learning (RL), where interac-

tions with the environment replace traditional text-based examples (Chen et al., 2021a; Xu et al., 2022; Laskin et al., 2023; Lee et al., 2023). In this paper, we focus on the robustness of in-context RL (ICRL) to reward poisoning attacks – one of the major security threats for safe deployment of RL.

Reward poisoning attacks have been extensively explored in recent RL literature (Lin et al., 2017; Ma et al., 2019; Zhang et al., 2020b; Wu et al., 2023; Nika et al., 2023). This line of work predominantly focuses on the canonical RL setting, modeling reward poisoning attacks as an attacker that corrupts the reward of a learning agent during training. In contrast to *test-time* adversarial attacks, poisoning attacks influence the policy that the agent adopts at test time; i.e., they are *training-time* attacks. This is perhaps not surprising, given that this line of work typically focuses on Markov stationary policies, implying that the agent's behavior is independent of the rewards at *test time*.

However, an ICRL agent can implement a *learning algorithm* in-context, using approaches such as Algorithm Distillation (Laskin et al., 2023) or the Decision-Pretrained Transformer (DPT, Lee et al., 2023). In this case, the context encodes past interactions (incl. rewards) between the environment and the agent. By corrupting the agent's rewards at test time, an adversary can still influence the agent's behavior. Simply put, such test-time reward poisoning schemes attack the learning algorithm implemented in-context.

In this work, we aim to develop a novel training protocol for ICRL that enables models to be robust against test-time reward poisoning attacks. We use DPT as a base approach. At a high level, it should implement a corruption-robust learning algorithm in-context. This differs from typical corruption-robust RL approaches (Lin et al., 2017; Ma et al., 2019; Zhang et al., 2020b; Sun et al., 2021; Wu et al., 2023; Nika et al., 2023): in our setting, rewards are corrupted only at test time, meaning the corruption does not affect the training process of the agent's policy. Our contributions are:

**Framework.** To our knowledge, we are the first to study reward poisoning in meta-RL – we introduce and formalize a novel attack modality predicated on reward poisoning. In this modality, the attacker influences the agent's context by corrupting the rewards. More specifically, the attacker aims to minimize the agent's return by modifying the reward under a soft budget constraint encoded via a penalty term.

†This work was done as a part of an internship project at MPI-SWS. [1]Department of Computing Science, University of Alberta, Edmonton, Canada. [2]Alberta Machine Intelligence Institute (Amii), Edmonton, Canada. [3]MPI-SWS, Saarbrücken, Germany. Correspondence to: Paulius Sasnauskas <sasnausk@ualberta.ca>.

*Proceedings of the $43^{rd}$ International Conference on Machine Learning*, Seoul, South Korea. PMLR 306, 2026. Copyright 2026 by the author(s).

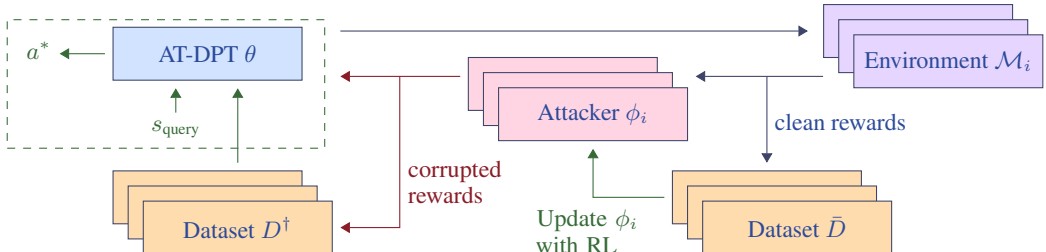

*Figure 1.* The training procedure of AT-DPT. We use adversarial training to optimize the parameters $\theta$ of a transformer model, which is our learning agent. In each round, we first collect data by deploying the agent in $m$ environments. In environment $i \in 0, \ldots, m$, the agent observes rewards corrupted by an adversary defined by parameters $\phi_i$. We collect clean ($\bar{D}$) and corrupted ($D^\dagger$) datasets containing trajectories with clean and corrupted rewards, respectively. The agent is trained to predict an optimal action for a query state given a context sampled from a corrupted dataset. The adversary is trained to minimize the agent's return under a soft budget constraint, expressed as a penalty term. During the test phase, the agent is deployed in a new corrupted environment.

**Method.** We combine in-context learning with adversarial training to develop an agent that is robust against reward poisoning. Specifically, we introduce the Adversarially Trained Decision-Pretrained Transformer (AT-DPT), which is trained by simultaneous optimization: a population of adversaries try to minimize the environment's reward, while DPT learns to infer optimal actions from the corrupted data. An overview of the training can be seen in Figure 1.

**Experiments.** We conduct a systematic evaluation of the proposed method, comparing its corruption-robustness capabilities to various methods, including robust baselines designed to handle reward contamination, under various levels of poisoning. Our results show that AT-DPT is robust against a wide range of reward poisoning attacks and yields better performance under corruption than the baselines.

**Importance.** Prior works on reward poisoning attacks primarily focus on single-environment training-time attacks, focusing on learning policies. In contrast, our work focuses on meta-RL, i.e., learning a learning algorithm itself. Hence, our goal is to provide a principled approach to designing learning algorithms inherently robust to corruptions. This can enable us to design more efficient algorithms that account for the distribution of tasks and utilize this information as a prior to obtain better performance. Our results show the potential of transformer-based policies in implementing algorithms that are robust against data contamination.

## 2. Related Work

**Adversarial ML.** Adversarial ML studies the effects of adversaries on models, as well as methods to defend against them. These adversaries have been extensively studied in computer vision, and recently many have been proposed in RL, including test-time attacks on observations (Biggio et al., 2013; Szegedy et al., 2014; Goodfellow et al., 2015; Papernot et al., 2017), (training-time) poisoning attacks

(Mei & Zhu, 2015; Li et al., 2016; Pattanaik et al., 2018), and backdoor attacks (Chen et al., 2017; Gu et al., 2017; Salem et al., 2022). Closest to our work are poisoning attacks on RL (Huang et al., 2017; Sun et al., 2021; Zhang et al., 2021), which have considered different targets, including transitions (Ma et al., 2019; Rakhsha et al., 2020), rewards (Lin et al., 2017; Zhang et al., 2020b; Sun et al., 2020; Wu et al., 2023; Nika et al., 2023), states (Zhang et al., 2020a), and actions (Rangi et al., 2022). Prior work also considered attacks in multi-agent RL (Mohammadi et al., 2023; Wu et al., 2024). We contribute to this area by studying the robustness of ICRL under *test-time* poisoning attacks.

**Corruption Robustness in RL.** Our work is closest to the works on corruption-robust bandits, RL, and multi-agent RL (Lykouris et al., 2018; Rakhsha et al., 2020; Niss & Tewari, 2020; Chen et al., 2021b; Lee et al., 2021; Wei et al., 2022; Ding et al., 2022; Nika et al., 2023; Xu et al., 2024). These works often establish guarantees for the suboptimality gap in terms of the level of corruption. Rather than focusing on theory, we contribute a practical method for training corruption robust ICRL. As explained in the introduction, this approach is conceptually different: corruption robust learning is implemented in-context. We experimentally compare the efficacy of our approach to bandit and RL algorithms robust to reward contamination, such as corruption robust UCB (Niss & Tewari, 2020; Ding et al., 2022) and Natural Policy Gradient (NPG, Kakade, 2001; Zhang et al., 2021).

Our work is also tied to the literature on robust offline RL (Yang et al., 2022; Panaganti et al., 2022; Ye et al., 2023; Yang et al., 2024; Yang & Xu, 2024). Prior work DeFog (Hu et al., 2023) or LHF (Chen et al., 2025) rely on filtering the learning histories during training. We note that both of these works are developed for robustness against random or noisy perturbations. For *adversarial* corruption robustness, another work (Xu et al., 2025) studies several improvements for the Decision Transformer. However, this method focuses

on the single-task setting, compared to ours.

**In-Context Reinforcement Learning.** Our research falls under ICRL, improving upon the initial DPT method (Lee et al., 2023). A similar work is Algorithm Distillation, which distills a policy which implicitly imitates policy improvement by training on episodic trajectories from learning algorithm histories (Laskin et al., 2023). Follow-up work by Zisman et al. (2024) involves injecting noise in the curriculum to allow generating learning histories without the need for optimal actions. However, both they and Dong et al. (2024) show that ICRL is sensitive to pretraining dataset perturbations. We also mention Tang et al. (2024), who study the Adversarially Robust Decision Transformer (ARDT) – a method robust against adaptive adversaries within a Markov game framework, capable of choosing actions which minimize the victim's rewards. This framework, translated to ours, would correspond to an adversary modifying transition probabilities and the victim observing the adversary's action. In contrast, we consider adversarial rewards generated by the attacker, and the victim only observes the realized reward without knowledge of whether an attacker is interfering, nor knowledge of their algorithm. For an in-depth discussion on ICRL, we refer to Moeini et al. (2025).

**Meta-RL.** Our work is broadly related to meta-RL, since we consider a multi-task setting. Meta-RL has been used in a variety of ways – optimizing a policy conditioned on histories of past transitions via an RNN (Duan et al., 2016), similarly, utilizing a Structured State Space Sequence model replacing the RNN (Lu et al., 2023), learning good *starting point* parameters that make learning in tasks faster (Finn et al., 2017), learning a dynamics model shared across tasks (Nagabandi et al., 2019). Transformers have also been utilized in prior work in learning multi-task policies (Reed et al., 2022; Lee et al., 2022). We refer the interested reader to Beck et al. (2025) for a detailed discussion on meta-RL.

**Other.** Recently there have been many works studying various different attacks on large language models (LLMs) to provoke an unsafe response (Zhao et al., 2024; He et al., 2025; Cheng et al., 2024, and many others), also called red-teaming (Ganguli et al., 2022). The increasing use of LLMs within decision-making systems provoke the need to study robustness. Within the text domain Cheng et al. (2024) find that pretraining a transformer with noisy labels robustifies it against that type of perturbation. Therefore, we advocate for the study of robust decision-making algorithms and hope our method contributes to this body of knowledge.

## 3. Setup

**Notation.** We will use $\Delta(\mathcal{A})$ to refer to the probability distribution over $\mathcal{A}$, and $\|\cdot\|_2$ denotes the Euclidean norm.

### 3.1. In-Context Sequential Decision-Making

**Environment.** We consider a multi-task sequential decision making setting, where we denote $\mathcal{T}$ as the distribution of tasks. Each task $\mathcal{M} \sim \mathcal{T}$ is formalized as an episodic finite-horizon Markov decision process (MDP) $\mathcal{M} = \langle \mathcal{S}, \mathcal{A}, R, T, H, \rho \rangle$, where $\mathcal{S}$ is the state space, $\mathcal{A}$ is the action space, $R : \mathcal{S} \times \mathcal{A} \to \Delta(\mathbb{R})$ is the reward function, $T : \mathcal{S} \times \mathcal{A} \to \Delta(\mathcal{S})$ is the transition function, $H \in \mathbb{N}$ is the horizon, and $\rho \in \Delta(\mathcal{S})$ is the starting state distribution. We denote realized states, actions, and rewards at timestep $h$ by $s_h$, $a_h$, and $r_h$, respectively. We distinguish between the clean and corrupted settings and use $\bar{r}_h$ to denote the true rewards, and $r_h^\dagger$ to denote the rewards produced by an attacker. The attack model is introduced in the next subsection. Let $\mu_{\bar{R}}(s, a)$ denote the mean of the underlying environment reward for the state-action pair $(s, a)$. For a stochastic policy $\pi : \mathcal{S} \to \Delta(\mathcal{A})$, the value function is defined as $V^\pi(\rho) = \mathbb{E}_{s \sim \rho} \left[ \sum_{h=1}^H \bar{r}_h \mid \pi, s_0 = s \right]$, where the expectation is w.r.t. the randomness of the underlying rewards when rolling out policy $\pi$ in $\mathcal{M}$. The solution to task $\mathcal{M}$ is an optimal policy $\pi_{\mathcal{M}}^\star$ that maximizes the value function, i.e., $V^{\pi_{\mathcal{M}}^\star}(\rho) = \max_\pi V^\pi(\rho)$.

**Agent.** We model a learning agent as a context-dependent policy parameterized by a transformer with parameters $\theta$ which maps the history of interactions $D$ and a query state $s_{\text{query}}$ to a distribution over actions. We denote this policy by $\pi_\theta(a_h \mid D, s_h)$ and $D = \{(s_i, a_i, r_i, s_{i+1})\}_{i=0}^{H-1}$ is the *in-context dataset* consisting of a set of previous interactions. To implement an efficient learner, we can train $\pi_\theta(\cdot \mid D, s_h)$ to predict optimal actions $a_h^\star \sim \pi_{\mathcal{M}}^\star(\cdot \mid s_h)$ for a task $\mathcal{M}$ sampled from a given task distribution – this approach is the backbone of the DPT (Lee et al., 2023).

### 3.2. Attack Model

We consider bounded reward poisoning attacks applied to a fraction of tasks at *test-time*. We employ Huber's $\varepsilon$-contamination model (Huber, 1964) and assume that the agent observes the corrupted reward in $\varepsilon$-fraction of timesteps. We model the attacker $\pi_\phi^\dagger : \mathcal{S} \times \mathcal{A} \times \mathbb{R} \times (\mathcal{S} \times \mathcal{A} \times \mathbb{R} \times \mathcal{S})^C \to \Delta(\mathbb{R})$ as a function of the state, action and reward of the last timestep along with an in-context dataset $\bar{D} \in (\mathcal{S} \times \mathcal{A} \times \mathbb{R} \times \mathcal{S})^C$ consisting of $C$ tuples of agent's interactions. Formally, at timestep $h$, the environment generates $\bar{r}_h \sim R(s_h, a_h)$, and the agent observes

$$\tilde{r}_h = \begin{cases} r_h^\dagger \sim \pi_\phi^\dagger(\cdot \mid s_h, a_h, \bar{r}_h, \bar{D}) & \text{with probability } \varepsilon, \\ \bar{r}_h & \text{otherwise.} \end{cases}$$

The attacker observes the underlying environment reward $\bar{r}_h$ to generate $r_h^\dagger$, but the victim $\pi_\theta$ only observes the realized reward $\tilde{r}_h$. We call an attacker *adaptive* if $C > 0$, meaning it leverages the agent's past interactions, and *non-adaptive*

if $C = 0$. In the non-adaptive case ($C = 0$) we simplify $\pi_\phi^\dagger(\cdot \mid s_h, a_h, \bar{r}_h, \bar{D}) = \pi_\phi^\dagger(\cdot \mid s_h, a_h, \bar{r}_h)$. In both cases the attacker aims to minimize the agent's expected return in $\mathcal{M}$ under a soft budget constraint, without forcing a specific policy for the agent. We denote the mean and variance of corrupted rewards by $\mu_\phi(s, a) = \mathbb{E}_{r^\dagger \sim \pi_\phi^\dagger(\cdot|s,a)}[r^\dagger]$ and $\sigma_\phi(s, a) = \mathrm{Var}_{r^\dagger \sim \pi_\phi^\dagger(\cdot|s,a)}[r^\dagger]$. The attacker's objective is

$$
\begin{aligned}
L(\mathcal{M}, \phi, \theta) = \mathbb{E}\left[\sum_{h=1}^{H} -\bar{r}_h \mid \pi_\theta, \pi_\phi^\dagger\right] \\
- \lambda \cdot c_\mu\left(\|\mu_\phi - \mu_{\bar{R}}\|_2\right) - \lambda \cdot c_\sigma\left(\|\sigma_\phi\|_2\right),
\end{aligned}
\tag{1}
$$

where we take the expectation over the stochasticity of the environment, the agent's policy, and the contamination model. $c_\mu$, $c_\sigma$ are penalty functions for exceeding budget $B$ and $B_\sigma$ respectively, and $\lambda > 0$ controls the strength. In both cases, we focus on non-behavior-targeted attacks (i.e., ones which do not force a specific policy), as opposed to behavior-targeted attacks, or policy-forcing attacks (Hussenot et al., 2020; Boloor et al., 2020).

### 3.3. In-Context RL With Corrupted Rewards

To account for the change induced by the attacker, we set the agent's objective to $U(\mathcal{M}, \theta, \phi) = \mathbb{E}\left[\sum_h^H \bar{r}_h \mid \pi_\theta, \pi_\phi^\dagger\right]$, where the expectation is taken over the randomness of the realized rewards when running the policy $\pi_\theta$ in $\mathcal{M}$, while corrupting its context $D^\dagger$ using the $\varepsilon$-contamination model with the attack policy $\pi_\phi^\dagger$.

We search for a Nash equilibrium $(\theta^\star, \{\phi_\mathcal{M}^\star\}_{\mathcal{M} \in \mathcal{T}})$ such that $\theta^\star \in \arg\max_\theta \mathbb{E}_{\mathcal{M} \in \mathcal{T}}[U(\mathcal{M}, \theta, \phi_\mathcal{M}^\star)]$ and $\phi_\mathcal{M}^\star \in \arg\max_\phi L(\mathcal{M}, \theta^\star, \phi)$ for all $\mathcal{M} \in \mathcal{T}$. To approximate this equilibrium empirically, we propose an adversarial training procedure. We sample $M$ tasks and optimize a dedicated attacker $\pi_{\phi_{\mathcal{M}_i}}^\dagger$ for each task $\mathcal{M}_i$, concurrently with the optimization of the agent $\pi_\theta$.

## 4. Method

We extend the work by Lee et al. (2023) with adversarial training to mitigate attacks which poison the context and influence future actions. The setup consists of three phases.

**Pretraining.** Lee et al. (2023) use GPT-2 as the underlying transformer model, and we adopt the same architecture. The model $\pi_\theta$ is initialized from scratch and trained via supervised learning by predicting an optimal action from context $D_{\text{pre}}$ and query state $s_q$. During the pretraining phase the model observes a dataset $D_{\text{pre}} \sim \mathcal{D}_{\text{pre}}$, which consists of tuples $(s, a, r, s')$ sampled from a set of $M$ tasks $\{\mathcal{M}_i \sim \mathcal{T}\}_{i=1}^M$. This dataset can be collected in various ways, e.g., random interactions with the environments. Alongside these interactions, we also sample a query

---

**Algorithm 1** Adversarial Training for AT-DPT

1: **input:** victim $\pi_\theta$ – DPT with pretrained params $\theta_0$
2: **input:** attacker $\pi_\phi^\dagger$ with initial params $\phi_0$, budget $B$, fraction of steps poisoned $\varepsilon$
3: Sample $M$ tasks $\{\mathcal{M}_i \sim \mathcal{T}\}_{i=1}^M$
4: **for** round $n$ in $0 \ldots N - 1$, simultaneously in all $\mathcal{M}$ **do**
5:     roll out $\pi_{\theta_n}$ for $H$ steps in $\mathcal{M}_i$ poisoned by $\pi_\phi^\dagger$ with $\varepsilon$-contamination model and budget $B$,
6:       where DPT collects corrupted dataset $D^\dagger$, and attacker collects dataset $\bar{D}$
7:     $\phi_{n+1} \leftarrow$ train on $\bar{D}$ with RL:
8:       see Equation (1)
9:     $\theta_{n+1} \leftarrow$ train on $D^\dagger$ via supervised learning:
10:       $\min_\theta \ell(\pi_\theta(\cdot \mid D^\dagger, s_q), a^\star)$, $a^\star$ provided by oracle
11: **end for**

---

state $s_q \sim \mathcal{D}_{\text{query}}$ and its corresponding optimal action $a^\star \sim \pi_\mathcal{M}^\star(\cdot \mid s_q)$. The model is then trained to minimize $\min_\theta \mathbb{E}_{D_{\text{pre}} \sim \mathcal{D}_{\text{pre}}, s_q \sim \mathcal{D}_{\text{query}}} \ell(\pi_\theta(\cdot|D_{\text{pre}}, s_q), a^\star)$, where $\ell$ is the NLL loss.

**In-Context Learning.** During the test phase $\pi_\theta$ is deployed in $\mathcal{M} \sim \mathcal{T}$ with an empty context $D = \{\}$. The original work updated the context $D$ with the entire trajectory $\{(s_h, a_h, r_h, s_{h+1})\}_{h=1}^H$ only after the entire episode (Lee et al., 2023). Whereas, in our method we update context $D$ with transitions $(s_h, a_h, r_h, s_{h+1})$ after every timestep $h$, to support robustness against adaptive attacks.

**Adversarial Training.** Before testing, we include an additional phase for adversarial training. An illustration of this training process is shown in Figure 1. In the adversarial setting $\pi_\theta$ is deployed in $\mathcal{M}$ under an attacker $\pi_\phi^\dagger$, contaminating the victim's dataset $D^\dagger$ as specified in the previous section. We account for this by introducing an additional adversarial training stage between the original pretraining and in-context learning. To train the agent and the attacker, recall that we use two different contexts – a context with poisoned rewards $D^\dagger$ for the agent, and a context with underlying rewards $\bar{D}$ for the attacker. We repeat this process for $N$ rounds, updating $\theta$ and $\phi$ after each round. Parameters $\theta$ are updated as in the original DPT setting, with $s_q$ sampled from the environment, and $a^\star$ provided by an oracle.[1] The pseudocode of this method can be found in Algorithm 1.

We consider attackers parameterized by $\phi$ (e.g., a neural network). To train the attacker we use the REINFORCE algorithm (Williams, 1992) – after each episode we update $\phi$ with the objective specified in Equation (1). Recall that while the victim $\pi_\theta$ only observes the realized reward $\tilde{r}_h$, the attacker has to have access to the underlying environ-

---

[1]In the algorithm we require access to clean environments sampled from $\mathcal{T}$ at training time, although offline trajectories could be used with simulated attacks and (near-)optimal actions.

ment reward $\bar{r}_h$. The attacker's goal is to poison a single algorithm, which we denote the **attacker target**. That is, a different policy might emerge from an attacker targeting DPT versus an attacker targeting TS.

**Bandit Setting.** In the bandit settings we consider a direct parameterization of a deterministic attack, i.e., for an action $a_h^{(i)}$ (choosing arm $i$) at timestep $h$ the attack is $\pi_\phi^\dagger(\cdot \mid a_h^{(i)}, \bar{r}_h) = \pi_\phi^\dagger(a_h^{(i)}, \bar{r}_h) = \bar{r}_h + \phi(i)$, where $\phi \in \mathbb{R}^{|\mathcal{A}|}$.

**Adaptive Attacker.** We also consider a context-dependent algorithm, e.g., a transformer, to enable the attacker to adapt to the defenses of the victim. For this we utilize the same architecture (GPT-2) as the victim. The interaction in the environment is then modified as follows. At the start of an episode empty context $\bar{D} = \{\}$ is initialized for the attacker. At every step $h$ the attacker samples a reward $r_h^\dagger \sim \pi_\phi^\dagger(\cdot \mid \bar{D}, s_h, a_h, \bar{r}_h)$ for the victim and appends $(s_h, a_h, \bar{r}_h)$ to the dataset $\bar{D}$.

**MDP Setting.** In the MDP setting we also consider a direct parameterization of a deterministic non-adaptive attack, i.e., for a state-action pair $(s^{(i)}, a^{(j)})$ the attack becomes $\pi_\phi^\dagger(\cdot \mid s^{(i)}, a^{(j)}, \bar{r}) = \pi_\phi^\dagger(s^{(i)}, a^{(j)}, \bar{r}) = \bar{r} + \phi(i, j)$, where $\phi \in \mathbb{R}^{|\mathcal{S}| \times |\mathcal{A}|}$.

## 5. Experiments

We sample $M = 200$ tasks to run in parallel. For each round we train both the attacker and DPT for multiple (e.g., 20) iterations on the same dataset. We set penalties for exceeding the budget $c_\mu(x; B) = \max(0, x - B)$ and $c_\sigma(x; B_\sigma) = \max(0, x - B_\sigma)$ with $\lambda = 10$. In all tables results are color coded using a gradient from orange (best) to white (worst), with color intensity decreasing as performance decreases (i.e., higher regret or lower reward).

### 5.1. Baseline Algorithms

To evaluate our method's performance in the bandit setting we compare it with widely used baselines: Thompson sampling (TS, Thompson, 1933), upper confidence bound (UCB, Auer et al., 2002), and corruption robust algorithms: robust Thompson sampling (RTS, Xu et al., 2024) – a TS-based algorithm robust to adversarial reward poisoning, which features an added term to the bonus term in TS, and corruption-robust upper confidence bound (crUCB, Niss & Tewari, 2020) – a UCB style algorithm robust to $\varepsilon$-contamination, where we chose the trimmed mean variant (the mean is estimated with a fraction of smallest and largest observed values removed for every arm). For linear bandits we compare our method to LinUCB (Li et al., 2010), and a corruption robust variant – CRLinUCB (Ding et al., 2022, Section 4).

For the MDP baselines we choose two policy-gradient based

methods – natural policy gradient (NPG, Kakade, 2001); proximal policy optimization (PPO, Schulman et al., 2017); and a value-based method – Q-learning (Watkins & Dayan, 1992). Additionally, we include DPT with frozen parameters (indicated as DPT) as a baseline to observe the effect of adversarial training, and we additionally include results with suboptimal demonstrations – rows indicated by AT-DPT (sub. 30%) mean a random 30% of timesteps the expert actions are replaced with the 2nd best arm. More details about the baselines can be found in Appendix A.

We note that the literature usually discusses theoretical algorithms which do not have practical implementations nor empirical results (Lykouris et al., 2018; 2021; Chen et al., 2021b). Given their principled nature it would be interesting to include them for comparison, but we believe it is outside the scope of this work to implement and tune them.

In addition to algorithm baselines, we also consider two baselines for evaluation. We show performance of the algorithms in the clean environment, and we also consider a uniform random attack – the poisoned reward for timestep $h$ is $r_h^\dagger = \bar{r}_h + \phi(i)$, where $\phi \in \mathbb{R}^{|\mathcal{A}|}$ is generated once at the start of evaluation by sampling from a uniform random distribution and clipped by the budget constraint $\|\phi\|_2 < B$.

### 5.2. Bandit Setting

**Environment.** We begin with empirical results in a simple scenario – the multi-armed bandit problem. We follow a similar bandit setup to that presented in the original DPT paper (Lee et al., 2023). We sample 5-armed bandits ($|\mathcal{A}| = 5$), each arm's reward function being a normal distribution $R(\cdot \mid s, a) = \mathcal{N}(\mu^{(a)}, \sigma^2)$, where $\mu^{(a)} \sim \text{Unif}[0, 1]$ independently and $\sigma = 0.3$. The optimal policy in this environment is to always choose the arm with the largest mean: $a^\star = \arg\max_a \mu^{(a)}$. We follow the same pretraining scheme as the original work. For evaluation, we present the empirical cumulative regret: $\sum_h \bar{r}(a^\star) - \bar{r}(a_h)$. Low regret indicates the policy is close to optimal.

**Hyperparameters** To pretrain DPT in the bandit setting we use the following architecture and hyperparameters. The Transformer has 4 layers, 4 attention heads per layer, embedding dimension – 32, no dropout. We set the context length equal to episode length $H = 500$, learning rate $\eta = 0.001$ and train for 400 epochs. We pretrain DPT in the same way as the original, see the work by Lee et al. (2023) for more details. For adversarial training we use $\eta = 0.0001$ for the victim and $\eta_{\text{attacker}} = 0.03$. We consider attackers with a diagonal covariance matrix, and set $B_\sigma = 1$.

**Adversarial training makes DPT robust to poisoning attacks.** In Figure 2, we present the training-time performance of DPT under adversarial training. The training curve shows the per-round cumulative regret, averaged across $M$ tasks

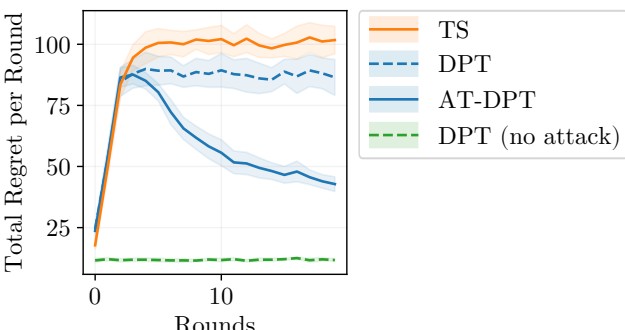

*Figure 2.* Comparison of the cumulative regret per round (lower is better) of different methods throughout 20 rounds of adversarial training (simultaneously learning AT-DPT and attackers) in the bandit setting. Within one round we perform $H = 500$ steps. The y axis indicates total regret for that round. Mean and 95% confidence interval ($2 \times$SEM) over 10 experiment replications. Attack budget $B = 3$, $\varepsilon = 0.4$.

seen during training. We observe the regret significantly increase in the first rounds, but in further rounds DPT learns to recover from the attacks, and results improve. In the figure we compare performance of TS and frozen DPT under the same attack, and also show the performance of frozen DPT on the clean environment (no attack). We refer to the adversarially trained DPT models as AT-DPT.

**Evaluation.** To evaluate AT-DPT on attacks trained for it, we cross-validate to prevent evaluation on the same attack AT-DPT has seen during training – we evaluate one AT-DPT with an attacker which is targeting AT-DPT for a different seed. We report the mean and 95% confidence interval ($2 \times$SEM) across 10 different experiment replications. During the test phase, AT-DPT uses trained parameters $\theta$, and the attacker uses $\phi$. The procedure for this can be seen in Appendix Algorithm 3. We note, that the tables and plots show the performance based on clean rewards, and not $\tilde{r}$. We run adversarial training for $N = 20$ rounds. Table 1 presents an extensive evaluation of AT-DPT and other method performance against attackers targeting different methods. We can clearly see AT-DPT outperforming all baselines in an

adversarial setting. Given that AT-DPT displays robustness against different attackers, it illustrates that it can successfully recover from attacks that are out-of-distribution.

**Adaptive Attacks.** For the adaptive attacker we utilize the same architecture as the victim, except without pretraining. In this setting we use $\eta_{\text{attacker}} = 0.00003$. Table 2 shows a comparison of performance under both adaptive and non-adaptive attackers. The table indicates that for AT-DPT the adaptive attacker is stronger. AT-DPT (A) performs better than AT-DPT (n-A) when evaluated against an adaptive attacker. This is unsurprising, but the performance difference is substantial and worth noting. There is a tradeoff: AT-DPT (n-A) performs better against non-adaptive attackers, though the difference is quite small. This mirrors the tradeoff observed comparing AT-DPT (n-A) to the baselines – it outperforms the baselines, but in the clean case it can slightly underperform. These tradeoffs are not surprising and simply suggest that it is useful to appropriately select the class of attacks expected in the target application.

### 5.3. Linear Bandit Setting

**Environment.** We follow a similar setup as in the original DPT work (Lee et al., 2023). We sample $d$-armed linear bandits, where the reward is given by $\mathbb{E}[r \mid a, \mathcal{M}_i] = \langle \omega_i, \psi(a) \rangle$, and $\omega_i \in \mathbb{R}^d$ is a task-specific parameter vector, and $\psi : \mathcal{A} \to \mathbb{R}^d$ is a feature vector shared across all tasks. Both $\omega_i$ for every $i$ and $\psi$ are sampled from $\mathcal{N}(\mathbf{0}, I_d/d)$. In our experiments we use $d = 2$ and $|\mathcal{A}| = 10$.

**Results.** From the results in Table 3 we see CRLinUCB performing only marginally better than all other algorithms in the clean case. Although, under a more complex attack AT-DPT outperforms all other algorithms, and matches CRLinUCB in the clean case.

### 5.4. MDP Setting

**Environment.** In the MDP setting we consider an extension of a sparse reward MDP considered in prior work – the Dark

*Table 1.* Comparison of cumulative regret (lower is better) of different algorithms under different attackers trained for 20 rounds, with $\varepsilon = 0.4$ steps poisoned. For $\varepsilon = 0.1$ and $0.2$ see Table 6 in the Appendix. Mean and 95% confidence interval ($2 \times$SEM) over 10 runs. Attack budget $B = 3$. [*] We use tuned versions of RTS and crUCB which outperform the base versions; full comparisons including base versions are given in Appendix A.

| Algorithm | Attacker Target | | | | | | Unif. Rand. Attack | Clean Env. |
| | AT-DPT | DPT | TS | RTS* | UCB1.0 | crUCB* | | |
|---|---|---|---|---|---|---|---|---|
| AT-DPT | $24.2 \pm 1.2$ | $24.8 \pm 1.4$ | $29.8 \pm 3.0$ | $28.3 \pm 1.9$ | $24.5 \pm 0.8$ | $23.8 \pm 1.4$ | $38.7 \pm 1.7$ | $13.0 \pm 0.9$ |
| AT-DPT (sub. 30%) | $41.2 \pm 2.9$ | $41.9 \pm 3.8$ | $45.7 \pm 3.3$ | $43.6 \pm 2.5$ | $41.6 \pm 3.0$ | $41.0 \pm 3.4$ | $47.8 \pm 3.5$ | $25.9 \pm 3.3$ |
| DPT | $63.6 \pm 8.6$ | $59.4 \pm 5.2$ | $62.0 \pm 8.6$ | $59.1 \pm 7.3$ | $55.4 \pm 8.1$ | $58.8 \pm 7.8$ | $37.2 \pm 1.2$ | $11.5 \pm 0.5$ |
| TS | $106.3 \pm 3.8$ | $97.7 \pm 4.9$ | $94.3 \pm 3.8$ | $93.1 \pm 6.0$ | $89.6 \pm 1.8$ | $92.6 \pm 4.8$ | $34.2 \pm 1.6$ | $8.7 \pm 0.6$ |
| RTS* | $102.9 \pm 4.5$ | $97.0 \pm 4.2$ | $90.4 \pm 4.4$ | $92.5 \pm 5.0$ | $89.2 \pm 3.4$ | $89.0 \pm 3.0$ | $33.9 \pm 1.6$ | $10.2 \pm 0.4$ |
| UCB1.0 | $104.1 \pm 3.6$ | $95.8 \pm 4.9$ | $90.6 \pm 4.1$ | $90.0 \pm 5.2$ | $88.1 \pm 3.4$ | $91.2 \pm 3.4$ | $38.1 \pm 2.2$ | $16.0 \pm 0.5$ |
| crUCB* | $86.0 \pm 4.4$ | $85.0 \pm 2.3$ | $82.0 \pm 4.4$ | $82.4 \pm 3.3$ | $79.4 \pm 3.0$ | $82.5 \pm 5.1$ | $31.8 \pm 1.6$ | $15.8 \pm 0.5$ |

*Table 2.* Comparison of the cumulative regret (lower is better) of adaptive and non-adaptive attackers. Attackers trained for 400 rounds, with $\varepsilon = 0.4$ steps poisoned. For $\varepsilon = 0.1$ and 0.2 see Table 8 in the Appendix. AT-DPT (A) means AT-DPT trained against the adaptive attacker, AT-DPT (n-A) means AT-DPT trained against the non-adaptive attacker. Mean and 95% confidence interval (2×SEM) over 10 replications. Attack budget $B = 3$. * We use tuned versions of RTS and crUCB which outperform base versions; details in Appendix A.

| Algorithm | Attacker Target | | | | Unif. Rand. | Clean Env. |
| | Adaptive | | Non-adaptive | | Attack | |
| | AT-DPT | TS | AT-DPT | TS | | |
|---|---|---|---|---|---|---|
| AT-DPT (A) | $37.1 \pm 6.6$ | $36.4 \pm 9.4$ | $38.0 \pm 6.4$ | $42.6 \pm 6.7$ | $41.4 \pm 7.3$ | $21.3 \pm 9.0$ |
| AT-DPT (n-A) | $88.1 \pm 20.0$ | $81.0 \pm 11.2$ | $22.8 \pm 1.6$ | $29.8 \pm 2.2$ | $39.7 \pm 3.8$ | $13.8 \pm 1.2$ |
| DPT | $97.9 \pm 18.6$ | $82.1 \pm 20.7$ | $61.6 \pm 8.0$ | $61.6 \pm 6.6$ | $37.3 \pm 3.5$ | $12.1 \pm 0.8$ |
| TS | $90.2 \pm 21.9$ | $104.2 \pm 26.7$ | $106.3 \pm 5.5$ | $94.3 \pm 4.8$ | $34.1 \pm 2.5$ | $9.1 \pm 0.7$ |
| RTS* | $90.5 \pm 21.3$ | $103.6 \pm 26.8$ | $104.5 \pm 5.5$ | $90.9 \pm 4.2$ | $34.5 \pm 2.4$ | $10.5 \pm 0.6$ |
| UCB | $94.3 \pm 22.4$ | $103.9 \pm 28.4$ | $101.3 \pm 5.0$ | $87.8 \pm 4.4$ | $38.2 \pm 1.6$ | $16.0 \pm 0.4$ |
| crUCB* | $85.1 \pm 23.5$ | $79.6 \pm 29.4$ | $88.4 \pm 4.4$ | $79.9 \pm 4.7$ | $32.0 \pm 1.7$ | $15.8 \pm 0.3$ |

Room environment (Lee et al., 2023; Laskin et al., 2023; Zintgraf et al., 2020) – a 2D gridworld environment where the agent only observes its own state and gains a reward of 1 when at the goal state. The agent has 5 actions – $\mathcal{A} = \{\text{up}, \text{down}, \text{left}, \text{right}, \text{stay}\}$. We consider a modification of this – instead of having one goal, we consider two goals – one giving a reward of 1, the other giving 2. To pretrain the DPT we supply optimal actions that lead to the goal giving reward of 2. We refer to this environment as Darkroom2.

To conform to the sparse reward nature of this environment we constrain the attacker to only output attacks in $\{-1, 0, 1\}$, having a softmax parameterization. This results in the observed reward being one of $\{-1, 0, 1, 2, 3\}$. We do not perform any reward normalization or scaling. In the evaluations we present the underlying episode reward $\sum_{h}^{H} \bar{r}_h$ as the performance metric. The budget of the attacker $B$ limits the norm of the vector of the attack at every cell.

**Hyperparameters** To pretrain DPT in the Darkroom2 setting we use the same model architecture as for the bandit setting, the context length equal to episode length $H = 200$, learning rate is $\eta = 0.0001$ and train for 150 epochs. For adversarial training we use the learning rate $\eta = 0.00003$ for the victim and $\eta_{\text{attacker}} = 0.03$.

**Evaluation.** To evaluate AT-DPT we perform cross-validation with different attackers, as in the bandit setting. For evaluation, we present the total underlying episode re-

ward $\sum_{h} \bar{r}_h$ in the tables. We report the mean and 95% confidence interval (2×SEM) across 10 different experiment replications. We run adversarial training for $N = 400$ rounds. The results, seen in Table 4, show that AT-DPT is robust against different attackers, with NPG coming close. The robustness displayed by NPG has been also observed by Zhang et al. (2021) – they find that NPG is robust against $\varepsilon$-contamination if the adversary rewards are bounded. We also observe that attacks with $\varepsilon = 0.1$ and $\varepsilon = 0.2$ are not very effective for NPG and Q-learning.

In Table 5 we additionally show experiments from the Miniworld environment (Chevalier-Boisvert et al., 2023), a 3D environment to evaluate visual navigation from images (25 × 25 pixels). We follow a similar setup as in the original DPT paper (Lee et al., 2023). The environment consists of four boxes of different colors, and one of those is chosen as the goal box, unknown to the agent. The agent receives a reward of $+1$ when stood next to the goal box. The episode is $H = 250$ steps long. These reseults show, that similarly to the smaller environments AT-DPT is more robust to adversarial reward corruption. That is, there is evidence to show that the method scales to larger environments.

The main advantage of using AT-DPT over classic algorithms in these scenarios is generalization – DPT is a meta-learner, which infers the task from a few interactions with the environment and follows an optimal policy almost immediately. Conversely, NPG and Q-learning are task-specific

*Table 3.* Comparison of the cumulative regret (lower is better) of the different algorithms under different attackers in the **linear bandit setting**, with $\varepsilon = 0.4$ steps poisoned. Attack budget $B = 3$. For $\varepsilon = 0.1$ and 0.2 see Table 9 in the Appendix. Mean and 95% confidence interval (2×SEM) over 10 replications. * We use a tuned version of CRLinUCB which outperforms the base version (see Appendix A.3).

| Algorithm | Attacker Target | | | | Unif. Rand. | Clean Env. |
| | AT-DPT | DPT | LinUCB | CRLinUCB* | Attack | |
|---|---|---|---|---|---|---|
| AT-DPT | $2.49 \pm 1.06$ | $2.50 \pm 1.08$ | $2.83 \pm 1.10$ | $1.79 \pm 1.02$ | $5.33 \pm 1.16$ | $3.89 \pm 0.86$ |
| DPT | $70.29 \pm 7.32$ | $71.42 \pm 7.46$ | $70.83 \pm 7.76$ | $63.84 \pm 7.18$ | $6.62 \pm 1.30$ | $3.35 \pm 0.84$ |
| LinUCB | $37.69 \pm 4.46$ | $35.93 \pm 3.86$ | $35.22 \pm 4.14$ | $34.82 \pm 4.36$ | $5.21 \pm 1.16$ | $3.51 \pm 0.88$ |
| CRLinUCB* | $37.45 \pm 4.76$ | $33.03 \pm 4.00$ | $35.56 \pm 4.26$ | $35.36 \pm 4.80$ | $5.12 \pm 1.48$ | $2.94 \pm 0.78$ |

*Table 4.* Comparison of the average episode reward (higher is better) of the different algorithms under different attackers trained for 300 rounds (5 rounds for Q-learning and NPG) in the **Darkroom2 environment** (5×5 grid). Mean and 95% confidence interval (2×SEM) over 10 replications, with $\varepsilon = 0.4$ steps poisoned. For $\varepsilon = 0.1$ and $0.2$ see Table 10 in the Appendix. Attack budget $B = 10$. [§] NPG and Q-learning need multiple episodes of learning to converge to a stable policy; we run them for 100 episodes before evaluating performance.

| Algorithm | Attacker Target | | | | Unif. Rand. Attack | Clean Env. |
|---|---|---|---|---|---|---|
| | AT-DPT | DPT | NPG | Q-learning | | |
| AT-DPT | $242.2 \pm 11.9$ | $267.5 \pm 10.5$ | $241.7 \pm 10.2$ | $239.1 \pm 8.8$ | $258.2 \pm 11.8$ | $267.4 \pm 15.1$ |
| AT-DPT (sub. 10%) | $225.2 \pm 17.8$ | $245.5 \pm 17.9$ | $226.1 \pm 16.6$ | $222.7 \pm 17.6$ | $241.3 \pm 17.8$ | $250.2 \pm 18.8$ |
| DPT | $216.1 \pm 11.0$ | $143.5 \pm 11.0$ | $202.6 \pm 7.4$ | $205.9 \pm 7.8$ | $266.2 \pm 8.1$ | $306.8 \pm 7.1$ |
| NPG[§] | $237.2 \pm 6.7$ | $243.7 \pm 7.9$ | $228.9 \pm 4.0$ | $228.1 \pm 8.1$ | $235.3 \pm 8.2$ | $241.7 \pm 7.5$ |
| Q-learning[§] | $198.1 \pm 3.7$ | $238.6 \pm 6.0$ | $215.4 \pm 7.6$ | $224.7 \pm 7.3$ | $229.0 \pm 7.2$ | $225.6 \pm 5.4$ |

learners – they require interactions from the current environment to improve their policies. In addition, these algorithms require a few (hundreds/thousands of) episodes before converging to a stable policy. In our experiments we trained different NPG, Q-learning, PPO policies for each environment. One could argue that it is possible to use a universal policy, but the agent is unaware what the current task is, thus it is unclear what it needs to condition on.

## 6. Discussion

**Significance.** The key contribution of this work is the setup itself – a set of provided ingredients which lead to test-time adversarial robustness, which has not been considered in prior work. This is done via simultaneously training a population of attackers minimizing the underlying environment rewards, and the victim optimizing for the optimal actions from the poisoned data. By showing extensive evaluations in the bandit and MDP setting we demonstrated that adversarial training can work for in-context RL, showing generalization across different attacks (incl. adaptive).

Our approach significantly outperforms baselines that do not account for the multi-task structure (e.g., Table 3, by more than 90%). We believe there are two reasons. First, the algorithm can use an informed prior over the distribution of tasks, making corruption schemes that are implausible according to the prior ineffective. Second, the algorithm

*Table 5.* Comparison of avg. episode reward (higher is better) of the algorithms under different attackers trained for 100 rounds in the **Miniworld environment** with $\varepsilon = 0.4$ steps poisoned. Mean and 95% confidence interval (2×SEM) over 10 runs. Attack budget $B = 5$. For $\varepsilon = 0.1$ and $0.2$ see Table 11 in the Appendix. [§] PPO needs multiple episodes of online learning to converge to a stable policy; we run it for 100 episodes before evaluating performance.

| Algorithm | Attacker Target | | Unif. Rand. Attack | Clean Env. |
|---|---|---|---|---|
| | AT-DPT | DPT | | |
| AT-DPT | $104.8 \pm 16.0$ | $116.8 \pm 18.8$ | $108.6 \pm 15.1$ | $112.7 \pm 23.9$ |
| DPT | $81.2 \pm 12.2$ | $70.2 \pm 15.0$ | $102.7 \pm 13.1$ | $110.0 \pm 14.7$ |
| PPO[§] | $83.5 \pm 7.4$ | $83.8 \pm 7.2$ | $92.9 \pm 7.3$ | $123.5 \pm 8.1$ |

can be thought of as posterior sampling that uses "corrected" posterior, which accounts for the corruption process.

The method introduced in this work is based on DPT (Lee et al., 2023), although extensions to other ICRL works could be possible, such as *Vintix* (Polubarov et al., 2025), or *ICPE* (Russo et al., 2026), which we leave for future work.

**Theoretical Insights.** At a high level, our approach can be viewed as approximately solving the bi-level objective: $\min_\theta \mathbb{E}_{\mathcal{M}_i, D \sim \mathcal{M}_i(\pi_\theta, \pi_{\phi,i}), s_q}[\ell(\pi_\theta(\cdot|s_q, D), a^*)]$ such that $\pi_{\phi,i} \in \mathrm{BR}(\mathcal{M}_i, \pi_\theta)$ for all tasks $\mathcal{M}_i$, where BR denotes the set of best responses of the adversary given the agent's policy, and $D \sim \mathcal{M}_i(\pi_\theta, \pi_{\phi,i})$ is collected under attacker $\pi_{\phi,i}$ and agent $\pi_\theta$, as specified in Section 3.2. AT-DPT's adversaries are trained to minimize the agent's return under a soft budget constraint, therefore the resulting policy $\pi_\theta$ is expected to perform robustly against a range of adversarial strategies. This is consistent with empirical results – we observe that AT-DPT's performance does not substantially degrade when evaluated against unseen adversaries. In equilibrium $(\theta^*, \phi^*)$ we have $\theta^* \in \arg\min_\theta \mathbb{E}_{\mathcal{M}_i, D \sim \mathcal{M}_i(\pi_\theta, \pi_{\phi^*,i}), s_q}[\ell(\pi_\theta(\cdot|s_q, D), a^*)]$. Theorem 1 of Lee et al. (2023) suggests that $\pi_{\theta^*}$ implements in-context posterior sampling, but with a *corrected* posterior that accounts for adversarial data corruption (i.e., $\pi_{\phi^*}$). As long as the underlying reward means remain identifiable under the corruption process, standard posterior sampling analyses are applicable. In this case, the regret of a learning algorithm implemented by $\pi_{\theta^*}$ depends on the extent of corruption since increased corruption that reduces reward separability leads to higher regret (by analogy with the lower bound by Lai & Robbins, 1985). Given that $\pi_{\phi^*}$ minimizes the agent's return (under a constraint) for $\pi_{\theta^*}$, the guarantees on $\pi_{\theta^*}$ hold when the test-time corruption differs from $\pi_{\phi^*}$ but satisfies the budget constraint. A complete theoretical account would require establishing that AT-DPT converges to an approximate equilibrium of the underlying game, and that the approximate nature of the resulting solution does not significantly affect the results. We believe that performing such an analysis is challenging and therefore leave it to future work.

**Practical Relevance.** In general, robustness to test-time poisoning attacks is important for agents relying on in-context learning. A prime example are LLM-based agents, whose context can be easily poisoned if they use external tools or RAG mechanisms, as demonstrated by prior work (Greshake et al., 2023; Carlini et al., 2024). Our work is generally important for understanding how to enable robustness to poisoning attacks for agents that exhibit ICRL capabilities – as shown by Monea et al. (2025), this includes LLMs. One of our goals was to provide a principled approach to learning a robust RL algorithm which better utilizes prior knowledge, which we can tailor to specific tasks of interest (if our prior is informed). This may yield significant performance gains, and our empirical results support this (e.g., see Table 3).

**Limitations and Future Work.** The main limitation of our method, also a limitation of DPT is the need of actions provided by the oracle for training (Lee et al., 2023). The authors of DPT propose relaxing this requirement by supplying actions generated by another RL agent which performs sufficiently well for the current task, but this might not be possible in an adversarial scenario. A different approach is viable – training on offline trajectories with simulated attackers. We provide experiments demonstrating the performance of AT-DPT if suboptimal demonstrations were given (*subopt.* rows in Tables 1 and 5) – AT-DPT allows for some percentage of suboptimal demonstrations, but the performance degrades when increasing this percentage.

We also observe in our results the capability of AT-DPT to generalize beyond the attack it has been trained on (i.e., adversarially trained against its own specific attacker, generalizes to an attacker trained for TS, for example). This suggests it may be possible to exploit this further by adversarially training AT-DPT with multiple different contamination levels $\varepsilon$. Additionally in our results we only consider a single attack specification per experiment. To make AT-DPT even more robust, and potentially alleviate the trade-off observed in the clean and random attack environment it would be possible to train AT-DPT with multiple different attack specifications (e.g., mixing in non-adaptive and adaptive attacks), which we leave as a direction for future work.

## Code Availability

Code available at github.com/PauliusSasnauskas/AT-DPT.

## Acknowledgements

This research was, in part, funded by the Deutsche Forschungsgemeinschaft (DFG, German Research Foundation) – project number 467367360. We acknowledge the use of computing resources generously provided by MPI-SWS.

## Impact Statement

Adversarial data poisoning attacks have potential to be used by malicious actors to generate attacks against real systems. Therefore, as is highlighted in the paper, it is important to study methods that are robust. Our work presents a strong case for robustifying against these kinds of attacks – it can enable us to design more efficient algorithms that account for the distribution of tasks and utilize this information as a prior to obtain better robustness. Data poisoning attacks have existed in the literature before, and have many potential societal consequences, none which we feel must be specifically highlighted here.

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

# Supplementary Material:Robust In-Context Reinforcement Learning Under Reward Poisoning Attacks

# A. Baseline algorithms

We believe a comparison with Algorithm Distillation (AD, Laskin et al., 2023) would be unfair to AD, because in contrast to DPT, AD is learning to imitate a learning algorithm. The two methods are conceptually different: AD distills an RL algorithm learning how to learn (i.e., predict an action by implicitly performing a policy improvement, given the previous steps), whereas DPT instead is trained on optimal actions (i.e., predict optimal action by implicitly understanding the goal, similarly to posterior sampling, given the previous steps). Furthermore, following the original setup of the AD paper it would be quite hard to collect training data given our attack model – there is no straightforward way to get trajectories displaying a policy improvement under an attacker. Regardless, we include a few experiments with AD in Table 6 for $\varepsilon = 0.4$.

## A.1. Robust TS

Xu et al. (2024) provide the Robust TS algorithm. This algorithm relies on a corruption level hyperparameter $\bar{C}$. The recommendation given by the authors is to set this to $\sum_h^H c_h \leq \bar{C}$, where $c_h$ is the corruption level (i.e., in our case, $c_h = r_h - \bar{r}_h$) for step $h$, if the corruption level is known. If the corruption level is unknown, the authors suggest setting $\bar{C} = \sqrt{H \frac{\ln |\mathcal{A}|}{|\mathcal{A}|}}$.

Following these recommendations, in an environment with $H = 500$, $|\mathcal{A}| = 5$, $\varepsilon = 0.4$, our preliminary findings are:

- assume corruption level is known: $\bar{C} \approx 120$ – RTS performance is worse than TS; indicated as RTS ($\bar{C}$ known);

- assume corruption level is unknown: $\bar{C} \approx 12.7$ – RTS performance is worse than TS; indicated as RTS ($\bar{C}$ unk.);

- tuned $\bar{C}$ for our setup: $\bar{C} = 0.5$ – RTS performance is better than TS; indicated as RTS ($\bar{C}$ tuned).

We report the best scores (obtained with the tuned variant) in the main text, giving the full three variant comparison in Table 6.

## A.2. crUCB

Niss & Tewari (2020) provide a few variants of the crUCB algorithm. We chose the $\alpha$-trimmed variant, which performs best empirically. We introduce a modification to the algorithm due to poor original variant empirical performance. The modified variant is shown in Algorithm 2, where $f$ – $\alpha$-trimmed mean function – if $n$ is the number of rewards observed for that arm, removes $\lceil \alpha n \rceil$ lowest and $\lceil \alpha n \rceil$ highest rewards observed for that specific arm; removing $2 \lceil \alpha n \rceil$ elements in total, and $\mathbf{x}_a^{(h)}$ – list of observed rewards for arm $a$ at step $h$.

---

**Algorithm 2** crUCB ($\alpha$-trimmed variant), modified

---

1: **input:** $\alpha$ – fraction of steps poisoned
2: **input:** $\sigma_0$ – upper bound on sub-Gaussian constant (hyperparameter)
3: **input:** $f$ – mean estimate function ($\alpha$-trimmed mean)
4: **for** step $h = 1, \ldots, H$ **do**
5:     **for** each $a \in \mathcal{A}$ **do**
6:        $\hat{\mu}_a^{(h)} \leftarrow f(\mathbf{x}_a^{(h)})$    ($\alpha$-trimmed mean estimate of rewards)
7:        $N_a^{(h)} \leftarrow$ number of times action $a$ has been played
8:     **end for**
9:     Choose action $a = \arg\max_{a \in \mathcal{A}} \hat{\mu}_a^{(h)} + \sigma_0 \left( \sqrt{\frac{4 \log(h)}{\left\lceil (1-2\alpha) N_a^{(h)} \right\rceil}} \right)$
10: **end for**

---

The original bonus term in the algorithm is $\frac{\sigma_0}{1 - 2\alpha} \left( \sqrt{\frac{4 \log(h)}{N_a^{(h)}}} \right)$.

Assume $f(\mathbf{z})$ with $n$ elements returns zero if $\mathbf{z}$ contains fewer than $n - 2 \lceil \alpha n \rceil$ elements. The failure is observed when the assumption above is true – the estimated mean returns zero, whereas the bonus is not infinity, leading to arms which have only been played one time have a very low score.

We report the best scores (obtained with the modified variant) in the main text, giving full results in Table 6 comparing:

- the original variant, indicated as crUCB (orig.) or (o.);

- the original variant with $\sigma_0$ scaled by $\sqrt{1-2\alpha}$, indicated as crUCB (low $\sigma_0$) or (l. $\sigma_0$);

- the modified variant, indicated as crUCB (mod.) or (m.).

### A.3. CRLinUCB

We source the CRLinUCB algorithm from Ding et al. (2022). The authors suggest setting the upper bound of the budget $C'$ to equal $\varepsilon BH$. We found that the algorithm did not perform well when set to this value. We then tuned this variant, and present a number of results in the tables:

- the original variant, denoted as CRLinUCBv1, where the hyperparameters are set to the values suggested by Theorem 1 by Ding et al. (2022);

- the variant where the bound is divided by the time horizon $H$, denoted as CRLinUCBv2, which approximately matches the values of the experiments of Ding et al. (2022);

- a third variant, CRLinUCBv3, where the hyperparameters are interpolated between v1 and v2, they are within the same order of magnitude with the geometric mean of the values used in v1 and v2.

In the main text we report the results from CRLinUCBv2, which seemed to work best in our case.

## B. Additional results

### B.1. Bandit setting

We present Table 6, which shows the full set of results from the bandit setting. Note, that RTS ($\bar{C}$ unk.), RTS ($\bar{C}$ known) did not perform well, as also noted in Appendix A.1. Similarly, crUCB (orig.) did not perform well, as noted in Appendix A.2. The high values obtained in the case, where the attacker is crUCB (o.) mean that the attacker trained against this algorithm was not performing well, and therefore led to a weak attack. Recall that the setup has a dual objective, and simply judging by the regret or reward of a single row or column is not enough.

We additionally include AT-DPT trained against a uniform random attack (indicated by AT-DPT (rand.)). The results for AT-DPT (rand.) show that training against random noise improves robustness in many cases, but is substantially worse than training against a strong adversary. This means for better robustness it is important to train a strong adversary.

We include experiments with Algorithm Distillation (AD, Laskin et al., 2023) for the case when $\varepsilon = 0.4$. We train AD with source algorithms TS (row *AD [TS]*), crUCB (m.) (row *AD [crUCB (m.)]*), and crUCB (m.) under corruption (row *AD [crUCB (m.) c.]*). All instances of AD were trained with embedding dimension 32, 4 layers, 4 attention heads, 0 dropout, 1e-4 learning rate, AdamW (with 1e-4 weight decay), batch size 64, but the experiment setup is otherwise the same as in the rest of the table. It seems that AD with a sufficiently good source algorithm (e.g., AD with source crUCB) does perform better than the source algorithm alone (e.g., just crUCB), which performs slightly better than simply DPT alone. Out of all of these AT-DPT still seems to have the lowest regret, which is expected, because it is trained to be robust to corruption. As mentioned previously, to train robustly for AD we would have to show policy improvement under corruption, which is difficult on its own; alternatively, supplying optimal actions to AD during adversarial training would just make it a DPT method.

*Table 6.* Comparison of the cumulative regret (lower is better) of the different algorithms under different attackers trained for 20 rounds, in the bandit setting. Mean and 95% confidence interval (2×SEM) over 10 experiment replications. Attack budget $B = 3$.

| Algorithm | Attacker Target | | | | | | | | Unif. Rand. Attack | Clean Env. |
|---|---|---|---|---|---|---|---|---|---|---|
| | AT-DPT | DPT | TS | RTS ($\bar{C}$ t.) | UCB1.0 | crUCB (o.) | crUCB (l. $\sigma_0$) | crUCB (m.) | | |
| | | | | | $\varepsilon = 0.1$ | | | | | |
| AT-DPT | $14.5 \pm 0.9$ | $13.9 \pm 0.7$ | $14.4 \pm 0.8$ | $14.6 \pm 1.0$ | $14.8 \pm 0.8$ | $14.4 \pm 1.4$ | $14.1 \pm 0.6$ | $14.2 \pm 1.3$ | $14.2 \pm 1.2$ | $12.4 \pm 1.1$ |
| AT-DPT (sub. 10%) | $21.0 \pm 1.6$ | $20.9 \pm 1.7$ | $21.0 \pm 1.4$ | $21.1 \pm 1.4$ | $21.1 \pm 1.6$ | $20.8 \pm 1.6$ | $20.6 \pm 1.6$ | $20.9 \pm 1.9$ | $19.5 \pm 2.0$ | $17.2 \pm 1.8$ |
| AT-DPT (sub. 20%) | $26.1 \pm 1.7$ | $26.0 \pm 1.9$ | $26.0 \pm 1.9$ | $26.0 \pm 1.7$ | $26.3 \pm 1.5$ | $26.0 \pm 1.7$ | $25.6 \pm 1.8$ | $25.6 \pm 2.0$ | $24.0 \pm 2.1$ | $21.9 \pm 2.2$ |
| AT-DPT (sub. 30%) | $32.1 \pm 2.3$ | $31.9 \pm 2.4$ | $32.0 \pm 2.2$ | $32.2 \pm 2.2$ | $32.2 \pm 2.2$ | $31.8 \pm 2.0$ | $31.7 \pm 2.3$ | $31.8 \pm 2.4$ | $29.1 \pm 2.3$ | $27.0 \pm 2.5$ |
| AT-DPT (rand.) | $18.4 \pm 1.2$ | $17.9 \pm 1.0$ | $17.4 \pm 1.0$ | $18.1 \pm 1.6$ | $17.9 \pm 0.9$ | $18.3 \pm 1.5$ | $17.4 \pm 1.2$ | $17.9 \pm 1.6$ | $14.6 \pm 1.1$ | $12.9 \pm 1.1$ |
| DPT | $24.2 \pm 1.8$ | $22.7 \pm 2.0$ | $22.1 \pm 1.3$ | $23.0 \pm 1.6$ | $22.2 \pm 1.6$ | $21.2 \pm 1.8$ | $22.6 \pm 1.5$ | $22.2 \pm 1.8$ | $15.2 \pm 0.8$ | $12.1 \pm 0.5$ |
| TS | $27.9 \pm 1.1$ | $26.6 \pm 2.1$ | $28.4 \pm 1.8$ | $27.2 \pm 1.6$ | $25.9 \pm 1.6$ | $22.4 \pm 2.1$ | $28.0 \pm 1.8$ | $28.6 \pm 1.8$ | $12.0 \pm 1.0$ | $8.9 \pm 0.4$ |
| RTS ($\bar{C}$ tuned) | $27.1 \pm 0.6$ | $27.0 \pm 1.4$ | $26.9 \pm 1.0$ | $26.5 \pm 2.0$ | $24.2 \pm 1.1$ | $21.3 \pm 1.4$ | $26.2 \pm 1.4$ | $26.8 \pm 1.2$ | $13.0 \pm 0.8$ | $10.5 \pm 0.3$ |
| RTS ($\bar{C}$ unk.) | $59.8 \pm 0.5$ | $59.4 \pm 0.8$ | $59.2 \pm 0.6$ | $59.7 \pm 0.7$ | $58.7 \pm 0.6$ | $55.7 \pm 1.0$ | $59.2 \pm 1.0$ | $59.1 \pm 0.7$ | $49.9 \pm 0.8$ | $49.3 \pm 0.8$ |
| RTS ($\bar{C}$ known) | $94.9 \pm 0.8$ | $94.4 \pm 0.9$ | $94.5 \pm 0.8$ | $94.6 \pm 1.2$ | $94.0 \pm 0.8$ | $91.8 \pm 1.1$ | $93.8 \pm 1.1$ | $94.4 \pm 1.0$ | $84.7 \pm 1.6$ | $84.1 \pm 1.7$ |
| UCB1.0 | $30.8 \pm 1.5$ | $28.5 \pm 1.5$ | $29.5 \pm 0.8$ | $28.5 \pm 1.8$ | $27.3 \pm 1.5$ | $24.5 \pm 1.0$ | $28.7 \pm 1.5$ | $29.4 \pm 1.3$ | $17.9 \pm 0.4$ | $16.1 \pm 0.3$ |
| crUCB (orig.) | $82.4 \pm 0.8$ | $81.9 \pm 0.7$ | $82.5 \pm 0.9$ | $81.9 \pm 1.3$ | $82.1 \pm 0.7$ | $81.3 \pm 1.0$ | $82.1 \pm 0.8$ | $81.9 \pm 0.8$ | $79.2 \pm 1.4$ | $79.3 \pm 1.5$ |
| crUCB (low $\sigma_0$) | $19.5 \pm 1.8$ | $19.1 \pm 1.2$ | $20.0 \pm 1.0$ | $18.8 \pm 2.1$ | $20.1 \pm 1.5$ | $18.6 \pm 1.7$ | $19.9 \pm 1.6$ | $19.6 \pm 2.0$ | $11.1 \pm 0.7$ | $9.3 \pm 0.5$ |
| crUCB (mod.) | $19.4 \pm 1.7$ | $18.4 \pm 1.2$ | $20.5 \pm 1.2$ | $17.8 \pm 1.6$ | $19.7 \pm 1.1$ | $18.7 \pm 1.6$ | $19.5 \pm 1.0$ | $18.4 \pm 1.6$ | $11.0 \pm 0.7$ | $9.2 \pm 0.3$ |
| | | | | | $\varepsilon = 0.2$ | | | | | |
| AT-DPT | $17.9 \pm 1.4$ | $17.1 \pm 1.4$ | $19.0 \pm 1.5$ | $18.4 \pm 1.3$ | $17.8 \pm 1.4$ | $17.4 \pm 1.9$ | $17.0 \pm 0.9$ | $16.9 \pm 1.2$ | $20.4 \pm 2.0$ | $14.5 \pm 1.8$ |
| AT-DPT (sub. 10%) | $24.7 \pm 1.3$ | $23.7 \pm 1.5$ | $25.9 \pm 1.9$ | $24.3 \pm 1.9$ | $24.0 \pm 1.5$ | $23.5 \pm 1.6$ | $23.6 \pm 1.2$ | $23.4 \pm 1.6$ | $24.7 \pm 1.7$ | $17.3 \pm 1.7$ |
| AT-DPT (sub. 20%) | $29.7 \pm 1.8$ | $30.1 \pm 2.0$ | $31.2 \pm 2.2$ | $30.4 \pm 2.1$ | $30.5 \pm 1.9$ | $29.3 \pm 1.7$ | $29.4 \pm 1.8$ | $29.3 \pm 1.8$ | $28.9 \pm 2.0$ | $22.0 \pm 2.1$ |
| AT-DPT (sub. 30%) | $35.6 \pm 2.1$ | $35.8 \pm 2.4$ | $36.8 \pm 2.2$ | $36.2 \pm 2.2$ | $35.5 \pm 2.4$ | $34.6 \pm 1.9$ | $35.0 \pm 2.3$ | $35.0 \pm 2.1$ | $33.8 \pm 2.1$ | $27.0 \pm 2.4$ |
| AT-DPT (rand.) | $23.9 \pm 2.1$ | $24.3 \pm 2.0$ | $24.8 \pm 2.5$ | $24.5 \pm 2.4$ | $24.1 \pm 2.3$ | $23.9 \pm 2.3$ | $23.6 \pm 2.5$ | $24.1 \pm 2.0$ | $21.0 \pm 2.0$ | $15.5 \pm 2.0$ |
| DPT | $35.2 \pm 4.1$ | $33.2 \pm 3.5$ | $37.1 \pm 5.2$ | $35.1 \pm 3.1$ | $33.0 \pm 3.5$ | $28.9 \pm 3.1$ | $35.1 \pm 3.7$ | $33.1 \pm 3.9$ | $22.3 \pm 1.1$ | $12.1 \pm 0.5$ |
| TS | $51.1 \pm 3.2$ | $48.6 \pm 3.5$ | $51.1 \pm 3.1$ | $50.1 \pm 2.7$ | $47.2 \pm 2.4$ | $33.7 \pm 3.3$ | $51.8 \pm 1.7$ | $49.9 \pm 3.8$ | $18.7 \pm 1.1$ | $8.9 \pm 0.4$ |
| RTS ($\bar{C}$ tuned) | $49.8 \pm 3.4$ | $44.8 \pm 2.7$ | $48.1 \pm 2.0$ | $48.3 \pm 3.1$ | $44.3 \pm 3.8$ | $32.4 \pm 2.5$ | $47.6 \pm 2.0$ | $46.3 \pm 1.7$ | $19.1 \pm 1.1$ | $10.5 \pm 0.3$ |
| RTS ($\bar{C}$ unk.) | $76.0 \pm 2.1$ | $73.2 \pm 1.2$ | $74.5 \pm 1.7$ | $74.2 \pm 1.7$ | $72.1 \pm 1.3$ | $63.5 \pm 1.2$ | $73.8 \pm 1.2$ | $73.7 \pm 1.2$ | $53.0 \pm 0.9$ | $49.3 \pm 0.8$ |
| RTS ($\bar{C}$ known) | $131.7 \pm 1.8$ | $130.7 \pm 1.5$ | $131.3 \pm 1.5$ | $130.9 \pm 1.5$ | $130.6 \pm 1.5$ | $127.0 \pm 1.6$ | $130.4 \pm 1.5$ | $131.0 \pm 1.6$ | $116.4 \pm 2.5$ | $115.7 \pm 2.6$ |
| UCB1.0 | $51.9 \pm 2.5$ | $46.7 \pm 1.8$ | $50.8 \pm 1.9$ | $47.3 \pm 1.9$ | $45.2 \pm 2.7$ | $34.2 \pm 2.3$ | $49.1 \pm 1.8$ | $48.1 \pm 2.7$ | $23.9 \pm 0.8$ | $16.1 \pm 0.3$ |
| crUCB (orig.) | $101.6 \pm 1.2$ | $100.8 \pm 1.2$ | $101.7 \pm 1.3$ | $100.5 \pm 1.6$ | $101.0 \pm 1.0$ | $98.8 \pm 1.3$ | $101.1 \pm 1.1$ | $101.3 \pm 1.2$ | $96.0 \pm 2.0$ | $95.6 \pm 2.0$ |
| crUCB (low $\sigma_0$) | $34.7 \pm 2.1$ | $33.6 \pm 2.1$ | $34.4 \pm 2.1$ | $31.6 \pm 2.8$ | $32.9 \pm 1.6$ | $30.5 \pm 3.0$ | $32.9 \pm 2.2$ | $34.1 \pm 1.9$ | $15.2 \pm 0.6$ | $9.4 \pm 0.6$ |
| crUCB (mod.) | $33.7 \pm 1.8$ | $33.6 \pm 1.7$ | $33.9 \pm 2.2$ | $31.1 \pm 2.3$ | $31.8 \pm 2.2$ | $29.9 \pm 3.0$ | $33.4 \pm 2.7$ | $33.5 \pm 1.5$ | $15.1 \pm 1.0$ | $9.5 \pm 0.3$ |
| | | | | | $\varepsilon = 0.4$ | | | | | |
| AT-DPT | $23.7 \pm 1.8$ | $24.1 \pm 2.0$ | $29.5 \pm 3.2$ | $26.9 \pm 1.8$ | $24.3 \pm 1.1$ | $23.6 \pm 2.0$ | $23.5 \pm 0.8$ | $23.4 \pm 1.7$ | $37.7 \pm 3.1$ | $13.2 \pm 0.7$ |
| AT-DPT (sub. 10%) | $30.6 \pm 2.3$ | $30.9 \pm 2.8$ | $35.9 \pm 3.7$ | $32.8 \pm 2.2$ | $30.9 \pm 2.4$ | $29.7 \pm 1.9$ | $29.9 \pm 1.8$ | $30.3 \pm 2.5$ | $41.6 \pm 2.4$ | $17.4 \pm 1.8$ |
| AT-DPT (sub. 20%) | $35.4 \pm 2.4$ | $35.8 \pm 3.2$ | $40.0 \pm 3.9$ | $37.9 \pm 2.4$ | $36.0 \pm 2.7$ | $35.1 \pm 3.3$ | $34.5 \pm 2.8$ | $34.7 \pm 2.9$ | $42.9 \pm 2.9$ | $21.0 \pm 2.9$ |
| AT-DPT (sub. 30%) | $41.2 \pm 2.9$ | $41.9 \pm 3.8$ | $45.7 \pm 3.3$ | $43.6 \pm 2.5$ | $41.6 \pm 3.0$ | $41.3 \pm 3.4$ | $42.0 \pm 2.1$ | $41.0 \pm 3.4$ | $47.8 \pm 3.5$ | $25.9 \pm 3.3$ |
| AT-DPT (rand.) | $39.3 \pm 4.1$ | $40.3 \pm 4.3$ | $43.4 \pm 3.7$ | $42.1 \pm 3.5$ | $40.0 \pm 2.7$ | $42.3 \pm 3.4$ | $40.6 \pm 3.9$ | $40.3 \pm 4.1$ | $38.3 \pm 3.5$ | $20.6 \pm 2.6$ |
| DPT | $65.1 \pm 8.0$ | $59.7 \pm 5.1$ | $62.0 \pm 8.5$ | $60.2 \pm 7.2$ | $55.3 \pm 7.8$ | $42.1 \pm 6.4$ | $58.2 \pm 7.1$ | $58.6 \pm 7.7$ | $35.7 \pm 3.7$ | $11.5 \pm 0.5$ |
| AD [crUCB (m.)] | $55.2 \pm 2.5$ | $56.1 \pm 1.5$ | $57.0 \pm 2.2$ | $53.9 \pm 1.3$ | $55.3 \pm 2.1$ | $50.5 \pm 2.1$ | $54.5 \pm 2.4$ | $52.7 \pm 2.0$ | $33.1 \pm 1.6$ | $16.1 \pm 0.5$ |
| AD [crUCB (m.) c.] | $83.9 \pm 4.6$ | $80.1 \pm 4.4$ | $77.6 \pm 3.8$ | $73.9 \pm 3.2$ | $76.1 \pm 3.4$ | $61.6 \pm 2.9$ | $76.2 \pm 3.7$ | $76.0 \pm 3.2$ | $32.9 \pm 1.8$ | $15.6 \pm 0.5$ |
| AD [TS] | $94.8 \pm 2.6$ | $84.5 \pm 4.7$ | $82.5 \pm 4.8$ | $75.0 \pm 5.3$ | $78.7 \pm 4.4$ | $46.3 \pm 2.4$ | $80.7 \pm 4.3$ | $81.6 \pm 1.8$ | $35.5 \pm 2.4$ | $9.1 \pm 0.6$ |
| TS | $107.1 \pm 2.4$ | $97.8 \pm 4.1$ | $94.1 \pm 3.8$ | $94.9 \pm 5.6$ | $90.8 \pm 2.4$ | $49.2 \pm 2.3$ | $92.1 \pm 6.5$ | $92.4 \pm 4.5$ | $34.4 \pm 3.1$ | $8.7 \pm 0.6$ |
| RTS ($\bar{C}$ tuned) | $105.8 \pm 2.4$ | $97.3 \pm 4.3$ | $90.4 \pm 4.4$ | $93.0 \pm 4.6$ | $89.9 \pm 3.9$ | $47.3 \pm 3.1$ | $90.1 \pm 5.1$ | $87.9 \pm 2.7$ | $33.7 \pm 2.0$ | $10.2 \pm 0.4$ |
| RTS ($\bar{C}$ unk.) | $113.9 \pm 2.1$ | $107.9 \pm 2.6$ | $104.3 \pm 3.0$ | $104.7 \pm 3.0$ | $103.3 \pm 2.5$ | $74.3 \pm 2.3$ | $103.9 \pm 2.7$ | $104.7 \pm 1.9$ | $62.3 \pm 1.4$ | $48.9 \pm 0.4$ |
| RTS ($\bar{C}$ known) | $156.4 \pm 1.9$ | $155.3 \pm 1.9$ | $154.6 \pm 1.7$ | $154.6 \pm 2.0$ | $154.7 \pm 2.0$ | $149.7 \pm 2.0$ | $154.9 \pm 1.9$ | $155.3 \pm 2.0$ | $139.7 \pm 1.9$ | $140.2 \pm 1.6$ |
| UCB1.0 | $105.6 \pm 3.0$ | $95.3 \pm 4.8$ | $91.1 \pm 4.2$ | $90.3 \pm 4.9$ | $88.7 \pm 3.2$ | $46.6 \pm 2.2$ | $91.8 \pm 4.0$ | $91.4 \pm 3.4$ | $39.2 \pm 2.1$ | $16.0 \pm 0.5$ |
| crUCB (orig.) | $148.6 \pm 1.9$ | $147.7 \pm 1.9$ | $147.9 \pm 1.9$ | $147.2 \pm 2.1$ | $147.8 \pm 1.7$ | $145.1 \pm 1.9$ | $147.5 \pm 1.8$ | $148.1 \pm 1.8$ | $141.2 \pm 2.0$ | $141.4 \pm 1.9$ |
| crUCB (low $\sigma_0$) | $86.4 \pm 2.7$ | $82.9 \pm 3.8$ | $82.5 \pm 3.1$ | $83.9 \pm 3.9$ | $80.9 \pm 4.1$ | $64.7 \pm 3.8$ | $82.5 \pm 4.0$ | $84.4 \pm 4.5$ | $31.1 \pm 1.5$ | $15.0 \pm 0.4$ |
| crUCB (mod.) | $85.7 \pm 3.7$ | $84.3 \pm 2.2$ | $81.9 \pm 4.4$ | $82.1 \pm 3.7$ | $79.3 \pm 2.9$ | $63.9 \pm 2.2$ | $80.1 \pm 3.7$ | $81.9 \pm 4.9$ | $31.8 \pm 1.4$ | $15.8 \pm 0.5$ |

## B.2. Different Budgets

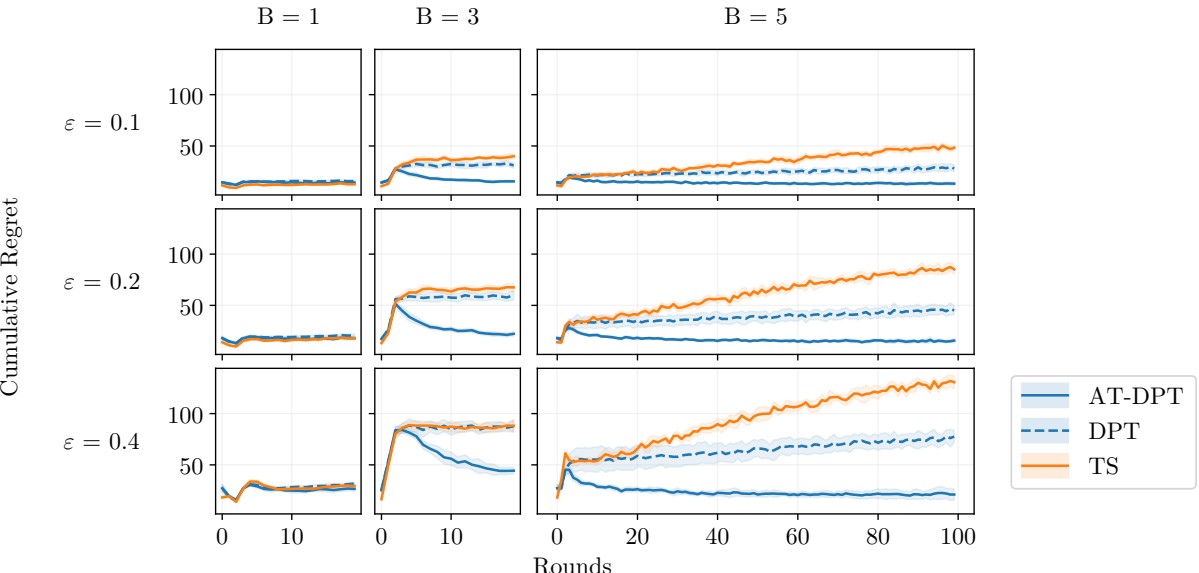

*Figure 3.* A study of the effect of the budget $B$ on the regret in the bandit setting. We run the experiments for $B = 5$ for more rounds to observe convergence. We observe that a larger budget for the attacker leads to a higher regret for TS and DPT, although adversarial training for AT-DPT helps it learn to recover from the attack.

*Table 7.* Comparison of the cumulative regret (lower is better) of the different algorithms under different attackers trained for 20 rounds, in the bandit setting, **across different budgets**. Mean and 95% confidence interval ($2\times$SEM) over 10 experiment replications. Fraction of steps poisoned $\varepsilon = 0.4$.

| Algorithm | | | Attacker Target | | | | Unif. Rand. Attack | Clean Env. |
|---|---|---|---|---|---|---|---|---|
| | AT-DPT | DPT | TS | RTS ($\bar{C}$ t.) | UCB1.0 | crUCB (m.) | | |
| $B_{\text{train}} = 1,\ B_{\text{test}} = 1$ | | | | | | | | |
| AT-DPT | $29.3 \pm 2.0$ | $28.2 \pm 2.2$ | $28.4 \pm 1.5$ | $28.6 \pm 1.8$ | $28.5 \pm 2.1$ | $28.5 \pm 2.5$ | $31.7 \pm 2.4$ | $14.7 \pm 1.9$ |
| DPT | $35.0 \pm 2.2$ | $31.6 \pm 1.7$ | $33.3 \pm 2.1$ | $32.8 \pm 1.6$ | $32.7 \pm 2.0$ | $33.0 \pm 3.1$ | $28.2 \pm 1.9$ | $11.5 \pm 0.5$ |
| TS | $34.3 \pm 2.0$ | $29.9 \pm 1.2$ | $30.1 \pm 1.0$ | $31.8 \pm 1.3$ | $30.3 \pm 1.7$ | $30.8 \pm 2.3$ | $26.7 \pm 2.2$ | $8.7 \pm 0.6$ |
| RTS ($\bar{C}$ tuned) | $34.3 \pm 1.9$ | $30.9 \pm 1.4$ | $30.6 \pm 1.8$ | $31.2 \pm 1.6$ | $30.2 \pm 0.9$ | $30.7 \pm 1.6$ | $27.1 \pm 1.6$ | $10.2 \pm 0.4$ |
| UCB1.0 | $40.1 \pm 1.6$ | $36.2 \pm 1.4$ | $37.2 \pm 1.2$ | $37.2 \pm 1.8$ | $36.6 \pm 1.7$ | $36.2 \pm 1.5$ | $31.4 \pm 1.5$ | $16.0 \pm 0.5$ |
| crUCB (mod.) | $37.1 \pm 1.4$ | $33.6 \pm 1.5$ | $34.3 \pm 1.5$ | $33.1 \pm 1.6$ | $33.3 \pm 1.8$ | $34.6 \pm 1.9$ | $26.9 \pm 0.9$ | $15.8 \pm 0.5$ |
| $B_{\text{train}} = 1,\ B_{\text{test}} = 3$ | | | | | | | | |
| AT-DPT | $28.5 \pm 1.6$ | $29.3 \pm 1.4$ | $34.1 \pm 2.9$ | $31.2 \pm 2.4$ | $28.9 \pm 2.1$ | $27.8 \pm 1.7$ | $38.5 \pm 6.4$ | $14.7 \pm 1.9$ |
| DPT | $63.5 \pm 6.6$ | $58.2 \pm 5.4$ | $61.4 \pm 7.5$ | $59.8 \pm 7.7$ | $57.4 \pm 8.1$ | $58.0 \pm 6.5$ | $34.1 \pm 5.5$ | $11.5 \pm 0.5$ |
| TS | $107.1 \pm 2.4$ | $97.8 \pm 4.1$ | $94.1 \pm 3.8$ | $94.9 \pm 5.6$ | $90.8 \pm 2.4$ | $92.4 \pm 4.5$ | $31.6 \pm 5.6$ | $8.7 \pm 0.6$ |
| RTS ($\bar{C}$ tuned) | $105.8 \pm 2.4$ | $97.3 \pm 4.3$ | $90.4 \pm 4.4$ | $93.0 \pm 4.6$ | $89.9 \pm 3.9$ | $87.9 \pm 2.7$ | $31.4 \pm 5.6$ | $10.2 \pm 0.4$ |
| UCB1.0 | $105.6 \pm 3.0$ | $95.3 \pm 4.8$ | $91.1 \pm 4.2$ | $90.3 \pm 4.9$ | $88.7 \pm 3.2$ | $91.4 \pm 3.4$ | $34.3 \pm 5.7$ | $16.0 \pm 0.5$ |
| crUCB (mod.) | $85.7 \pm 3.7$ | $84.3 \pm 2.2$ | $81.9 \pm 4.4$ | $82.1 \pm 3.7$ | $79.3 \pm 2.9$ | $81.9 \pm 4.9$ | $28.9 \pm 3.6$ | $15.8 \pm 0.5$ |
| $B_{\text{train}} = 1,\ B_{\text{test}} = 5$ | | | | | | | | |
| AT-DPT | $34.9 \pm 1.7$ | $35.7 \pm 2.3$ | $40.5 \pm 3.5$ | $37.7 \pm 2.6$ | $35.5 \pm 2.1$ | $35.0 \pm 2.9$ | $69.8 \pm 12.6$ | $14.7 \pm 1.9$ |
| DPT | $58.7 \pm 8.4$ | $56.1 \pm 11.2$ | $61.7 \pm 9.3$ | $59.2 \pm 10.0$ | $57.1 \pm 9.7$ | $56.6 \pm 10.6$ | $58.3 \pm 10.0$ | $11.5 \pm 0.5$ |
| TS | $65.0 \pm 5.5$ | $59.5 \pm 6.1$ | $65.3 \pm 4.1$ | $60.9 \pm 6.1$ | $53.7 \pm 3.8$ | $59.8 \pm 4.3$ | $54.9 \pm 10.4$ | $8.7 \pm 0.6$ |
| RTS ($\bar{C}$ tuned) | $61.5 \pm 5.3$ | $57.9 \pm 4.9$ | $62.3 \pm 5.3$ | $61.2 \pm 6.1$ | $53.1 \pm 4.7$ | $56.8 \pm 3.7$ | $53.4 \pm 10.6$ | $10.2 \pm 0.4$ |
| UCB1.0 | $60.1 \pm 4.6$ | $53.2 \pm 4.5$ | $64.4 \pm 4.4$ | $58.5 \pm 5.5$ | $50.5 \pm 3.9$ | $52.2 \pm 3.9$ | $59.5 \pm 12.1$ | $16.0 \pm 0.5$ |
| crUCB (mod.) | $79.9 \pm 2.9$ | $80.2 \pm 3.6$ | $82.0 \pm 4.4$ | $81.6 \pm 3.2$ | $75.3 \pm 5.1$ | $80.0 \pm 3.4$ | $41.9 \pm 6.5$ | $15.8 \pm 0.5$ |

We present experiments showing how AT-DPT trains on different attack budgets $B$ in Figure 3. In all three variants tested ($B = 1, 3, 5$) AT-DPT seems to recover within approx. 20 rounds and performs well.

While we test on out-of-distribution attacks in all experiments, meaning the test-phase attacks differ from the train-phase attacks, we perform an orthogonal out-of-distribution study on the impact of the budget during training ($B_{\text{train}}$) and evaluation ($B_{\text{test}}$) in Table 7. This shows how training on a weaker attacker ($B = 1$) transfers to stronger attacks ($B = 3$, $B = 5$). The setup is otherwise the same as in the other tables. The results seem to show that there is evidence that AT-DPT transfers robustness to stronger attacks. That is, AT-DPT trained with a weaker type of attack learns to be robust against a stronger attacker. We think it would be interesting to consider generalization to different attack vectors, such as observation or action poisoning, in future work, and there is potential to apply an approach similar to ours.

### B.3. Bandit setting, adaptive attack

Table 8 presents the full set of results comparing adaptive and non-adaptive attacks. The adaptive attacker columns in the table highlight, that these attacks work much better from the attacker's perspective, i.e., the attacks increase regret by a larger margin than in the non-adaptive case. We note that in both cases the regret of AT-DPT is low, meaning it is working well.

*Table 8.* Comparison of the cumulative regret (lower is better) of **adaptive and non-adaptive attackers** in the bandit setting. Attackers trained for 400 rounds. Mean and 95% confidence interval (2×SEM) over 10 experiment replications. Attack budget $B = 3$. * We use tuned versions of RTS and crUCB which outperform the base versions – see Appendix A for details.

| Algorithm | Adaptive | | Non-adaptive | | Unif. Rand. Attack | Clean Env. |
| --- | --- | --- | --- | --- | --- | --- |
| | AT-DPT | TS | AT-DPT | TS | | |
| $\varepsilon = 0.1$ | | | | | | |
| AT-DPT (against adaptive) | $21.5 \pm 4.2$ | $22.6 \pm 5.5$ | $19.6 \pm 3.8$ | $19.6 \pm 4.2$ | $21.4 \pm 9.4$ | $16.7 \pm 5.1$ |
| AT-DPT (against non-adaptive) | $44.6 \pm 17.7$ | $39.7 \pm 15.4$ | $14.1 \pm 0.7$ | $14.2 \pm 0.6$ | $14.7 \pm 1.0$ | $11.8 \pm 1.1$ |
| DPT | $60.2 \pm 18.5$ | $47.1 \pm 10.1$ | $22.2 \pm 1.1$ | $21.9 \pm 1.5$ | $15.1 \pm 1.0$ | $12.1 \pm 0.8$ |
| TS | $102.8 \pm 27.1$ | $91.0 \pm 29.3$ | $27.8 \pm 1.7$ | $27.4 \pm 1.6$ | $11.9 \pm 0.7$ | $9.1 \pm 0.7$ |
| RTS ($\bar{C}$ tuned) | $101.5 \pm 26.2$ | $91.7 \pm 29.0$ | $26.1 \pm 2.1$ | $26.4 \pm 2.1$ | $12.9 \pm 0.7$ | $10.5 \pm 0.6$ |
| UCB | $103.9 \pm 26.0$ | $94.0 \pm 27.2$ | $28.8 \pm 1.1$ | $29.2 \pm 1.1$ | $18.1 \pm 0.6$ | $16.0 \pm 0.4$ |
| crUCB (mod.) | $67.2 \pm 17.1$ | $53.3 \pm 15.8$ | $18.9 \pm 1.3$ | $19.6 \pm 1.5$ | $10.8 \pm 0.5$ | $9.2 \pm 0.3$ |
| $\varepsilon = 0.2$ | | | | | | |
| AT-DPT (against adaptive) | $26.6 \pm 5.3$ | $29.1 \pm 8.5$ | $25.9 \pm 5.1$ | $27.0 \pm 4.5$ | $27.2 \pm 10.9$ | $18.8 \pm 7.6$ |
| AT-DPT (against non-adaptive) | $54.7 \pm 15.5$ | $51.8 \pm 11.8$ | $17.5 \pm 0.9$ | $19.4 \pm 1.3$ | $20.5 \pm 1.9$ | $12.4 \pm 1.3$ |
| DPT | $71.3 \pm 20.4$ | $61.6 \pm 17.0$ | $34.6 \pm 3.6$ | $35.1 \pm 3.1$ | $22.0 \pm 1.9$ | $12.1 \pm 0.8$ |
| TS | $74.9 \pm 24.6$ | $91.6 \pm 35.0$ | $51.6 \pm 2.6$ | $51.8 \pm 3.6$ | $18.8 \pm 1.5$ | $9.1 \pm 0.7$ |
| RTS ($\bar{C}$ tuned) | $75.5 \pm 24.6$ | $92.2 \pm 34.2$ | $49.5 \pm 2.9$ | $49.3 \pm 2.7$ | $19.3 \pm 1.2$ | $10.5 \pm 0.6$ |
| UCB | $76.8 \pm 22.4$ | $92.4 \pm 30.9$ | $51.6 \pm 2.7$ | $49.4 \pm 1.9$ | $23.6 \pm 1.1$ | $16.0 \pm 0.4$ |
| crUCB (mod.) | $55.1 \pm 16.5$ | $61.8 \pm 25.0$ | $34.6 \pm 1.8$ | $34.4 \pm 1.2$ | $14.1 \pm 0.8$ | $9.5 \pm 0.5$ |
| $\varepsilon = 0.4$ | | | | | | |
| AT-DPT (against adaptive) | $37.1 \pm 6.6$ | $36.4 \pm 9.4$ | $38.0 \pm 6.4$ | $42.6 \pm 6.7$ | $41.4 \pm 7.3$ | $21.3 \pm 9.0$ |
| AT-DPT (against non-adaptive) | $88.1 \pm 20.0$ | $81.0 \pm 11.2$ | $22.8 \pm 1.6$ | $29.8 \pm 2.2$ | $39.7 \pm 3.8$ | $13.8 \pm 1.2$ |
| DPT | $97.9 \pm 18.6$ | $82.1 \pm 20.7$ | $61.6 \pm 8.0$ | $61.6 \pm 6.6$ | $37.3 \pm 3.5$ | $12.1 \pm 0.8$ |
| TS | $90.2 \pm 21.9$ | $104.2 \pm 26.7$ | $106.3 \pm 5.5$ | $94.3 \pm 4.8$ | $34.1 \pm 2.5$ | $9.1 \pm 0.7$ |
| RTS ($\bar{C}$ tuned) | $90.5 \pm 21.3$ | $103.6 \pm 26.8$ | $104.5 \pm 5.5$ | $90.9 \pm 4.2$ | $34.5 \pm 2.4$ | $10.5 \pm 0.6$ |
| UCB | $94.3 \pm 22.4$ | $103.9 \pm 28.4$ | $101.3 \pm 5.0$ | $87.8 \pm 4.4$ | $38.2 \pm 1.6$ | $16.0 \pm 0.4$ |
| crUCB (mod.) | $85.1 \pm 23.5$ | $79.6 \pm 29.4$ | $88.4 \pm 4.4$ | $79.9 \pm 4.7$ | $32.0 \pm 1.7$ | $15.8 \pm 0.3$ |

## B.4. Linear bandit setting

Table 9 presents the full results from the linear bandit setting. As described in Appendix A.3, CRLinUCBv1 and CRLin-UCBv3 performed worse than the tuned version CRLinUCBv2. This is indicated by their poor performance on the clean and uniform random attack cases.

*Table 9.* Comparison of the cumulative regret (lower is better) of the different algorithms under different attackers in the **linear bandit setting**. Mean and 95% confidence interval (2×SEM) over 10 experiment replications. Attack budget $B = 3$.

| Algorithm | Attacker Target | | | | | | Unif. Rand. Attack | Clean Env. |
|---|---|---|---|---|---|---|---|---|
| | AT-DPT | DPT | LinUCB | CRLinUCBv1 | CRLinUCBv2 | CRLinUCBv3 | | |
| $\varepsilon = 0.1$ | | | | | | | | |
| AT-DPT | $2.55 \pm 0.88$ | $2.02 \pm 0.92$ | $2.28 \pm 0.90$ | $2.44 \pm 0.96$ | $1.55 \pm 0.98$ | $1.85 \pm 0.94$ | $4.60 \pm 0.98$ | $3.89 \pm 0.86$ |
| DPT | $14.50 \pm 2.30$ | $14.42 \pm 2.48$ | $14.62 \pm 2.34$ | $14.06 \pm 2.72$ | $14.02 \pm 2.74$ | $13.19 \pm 2.36$ | $5.23 \pm 1.02$ | $3.35 \pm 0.84$ |
| LinUCB | $10.57 \pm 2.00$ | $7.92 \pm 1.24$ | $7.56 \pm 1.20$ | $9.65 \pm 1.78$ | $8.04 \pm 1.52$ | $9.23 \pm 1.72$ | $4.18 \pm 0.90$ | $3.51 \pm 0.88$ |
| CRLinUCBv1 | $104.00 \pm 7.06$ | $103.24 \pm 7.00$ | $103.11 \pm 7.04$ | $103.03 \pm 6.98$ | $102.56 \pm 7.02$ | $102.97 \pm 7.00$ | $102.95 \pm 7.00$ | $110.85 \pm 7.78$ |
| CRLinUCBv2 | $7.85 \pm 1.58$ | $7.99 \pm 1.60$ | $10.16 \pm 2.00$ | $10.94 \pm 2.34$ | $7.82 \pm 1.92$ | $9.07 \pm 2.48$ | $3.16 \pm 0.96$ | $2.94 \pm 0.78$ |
| CRLinUCBv3 | $17.66 \pm 1.18$ | $17.34 \pm 1.18$ | $17.40 \pm 1.24$ | $17.86 \pm 1.20$ | $16.99 \pm 1.14$ | $16.58 \pm 1.12$ | $13.55 \pm 0.96$ | $34.08 \pm 1.66$ |
| $\varepsilon = 0.2$ | | | | | | | | |
| AT-DPT | $1.37 \pm 0.94$ | $1.20 \pm 0.92$ | $1.67 \pm 0.96$ | $1.30 \pm 0.96$ | $2.42 \pm 0.94$ | $2.00 \pm 0.92$ | $4.80 \pm 0.98$ | $3.89 \pm 0.86$ |
| DPT | $33.65 \pm 4.14$ | $35.49 \pm 4.66$ | $32.23 \pm 3.96$ | $35.29 \pm 4.24$ | $33.67 \pm 4.02$ | $33.15 \pm 3.84$ | $5.91 \pm 1.12$ | $3.35 \pm 0.84$ |
| LinUCB | $19.11 \pm 2.56$ | $15.79 \pm 2.32$ | $18.80 \pm 2.62$ | $21.31 \pm 3.16$ | $16.95 \pm 2.68$ | $19.52 \pm 2.62$ | $4.37 \pm 0.98$ | $3.51 \pm 0.88$ |
| CRLinUCBv1 | $100.19 \pm 6.88$ | $99.73 \pm 6.74$ | $99.35 \pm 6.82$ | $99.41 \pm 6.88$ | $100.48 \pm 6.90$ | $100.34 \pm 6.88$ | $107.34 \pm 7.44$ | $110.85 \pm 7.78$ |
| CRLinUCBv2 | $16.42 \pm 2.76$ | $13.97 \pm 2.46$ | $16.56 \pm 2.46$ | $22.27 \pm 3.66$ | $16.02 \pm 2.64$ | $18.45 \pm 2.78$ | $3.41 \pm 1.02$ | $2.94 \pm 0.78$ |
| CRLinUCBv3 | $31.53 \pm 1.76$ | $28.93 \pm 1.62$ | $28.66 \pm 1.58$ | $30.44 \pm 1.80$ | $30.02 \pm 1.64$ | $30.20 \pm 1.74$ | $19.42 \pm 1.08$ | $34.08 \pm 1.66$ |
| $\varepsilon = 0.4$ | | | | | | | | |
| AT-DPT | $2.49 \pm 1.06$ | $2.50 \pm 1.08$ | $2.83 \pm 1.10$ | $2.93 \pm 1.06$ | $1.79 \pm 1.02$ | $2.16 \pm 1.10$ | $5.33 \pm 1.16$ | $3.89 \pm 0.86$ |
| DPT | $70.29 \pm 7.32$ | $71.42 \pm 7.46$ | $70.83 \pm 7.76$ | $69.49 \pm 6.88$ | $63.84 \pm 7.18$ | $73.45 \pm 6.82$ | $6.62 \pm 1.30$ | $3.35 \pm 0.84$ |
| LinUCB | $37.69 \pm 4.46$ | $35.93 \pm 3.86$ | $35.22 \pm 4.14$ | $49.39 \pm 5.12$ | $34.82 \pm 4.36$ | $39.97 \pm 4.50$ | $5.21 \pm 1.16$ | $3.51 \pm 0.88$ |
| CRLinUCBv1 | $108.12 \pm 6.96$ | $107.54 \pm 7.00$ | $108.04 \pm 6.98$ | $107.84 \pm 6.98$ | $106.79 \pm 6.98$ | $107.13 \pm 6.90$ | $109.46 \pm 7.64$ | $110.85 \pm 7.78$ |
| CRLinUCBv2 | $37.45 \pm 4.76$ | $33.03 \pm 4.00$ | $35.56 \pm 4.26$ | $46.23 \pm 5.46$ | $35.36 \pm 4.80$ | $37.75 \pm 4.80$ | $5.12 \pm 1.48$ | $2.94 \pm 0.78$ |
| CRLinUCBv3 | $53.23 \pm 3.02$ | $51.76 \pm 2.84$ | $53.31 \pm 2.98$ | $54.45 \pm 3.08$ | $49.13 \pm 2.66$ | $51.43 \pm 2.78$ | $28.34 \pm 1.56$ | $34.08 \pm 1.66$ |

## B.5. MDP setting

*Table 10.* Comparison of the average episode reward (higher is better) of the different algorithms under different attackers trained for 300 rounds (5 rounds for Q-learning and NPG) in the **Darkroom2 environment** ($5 \times 5$ grid). Mean and 95% confidence interval ($2 \times$ SEM) over 10 experiment replications. Attack budget $B = 10$. $\S$ NPG and Q-learning require multiple episodes of online learning to converge to a stable policy; we run them for 100 episodes before evaluating their performance.

| Algorithm | Attacker Target | | | | Unif. Rand. Attack | Clean Env. |
|---|---|---|---|---|---|---|
| | AT-DPT | DPT | NPG | Q-learning | | |
| $\varepsilon = 0.1$ | | | | | | |
| AT-DPT | $269.9 \pm 16.3$ | $266.0 \pm 20.2$ | $262.3 \pm 16.6$ | $258.9 \pm 20.2$ | $271.4 \pm 20.1$ | $272.7 \pm 18.3$ |
| AT-DPT (sub. 10%) | $262.0 \pm 23.1$ | $267.6 \pm 22.9$ | $261.3 \pm 24.3$ | $263.3 \pm 25.5$ | $271.6 \pm 22.7$ | $274.8 \pm 21.4$ |
| AT-DPT (sub. 20%) | $248.4 \pm 21.0$ | $246.6 \pm 19.6$ | $244.4 \pm 19.8$ | $242.5 \pm 22.3$ | $249.2 \pm 23.0$ | $247.1 \pm 21.7$ |
| AT-DPT (sub. 30%) | $217.1 \pm 18.0$ | $218.8 \pm 18.4$ | $213.3 \pm 18.3$ | $215.6 \pm 17.1$ | $222.1 \pm 18.9$ | $221.6 \pm 16.5$ |
| DPT | $236.8 \pm 9.7$ | $199.0 \pm 10.4$ | $224.8 \pm 12.4$ | $222.6 \pm 6.8$ | $277.4 \pm 7.1$ | $306.8 \pm 7.1$ |
| NPG$^\S$ | $241.9 \pm 6.6$ | $248.1 \pm 6.3$ | $247.7 \pm 7.1$ | $243.3 \pm 5.5$ | $246.0 \pm 6.9$ | $241.7 \pm 7.5$ |
| Q-learning$^\S$ | $280.1 \pm 5.5$ | $281.1 \pm 5.2$ | $248.5 \pm 38.6$ | $264.1 \pm 18.1$ | $266.5 \pm 15.4$ | $266.0 \pm 14.8$ |
| $\varepsilon = 0.2$ | | | | | | |
| AT-DPT | $261.0 \pm 14.6$ | $271.7 \pm 15.5$ | $257.7 \pm 16.7$ | $258.0 \pm 19.3$ | $270.1 \pm 17.8$ | $279.9 \pm 20.0$ |
| AT-DPT (sub. 10%) | $252.9 \pm 15.4$ | $259.8 \pm 12.3$ | $253.9 \pm 16.1$ | $255.3 \pm 14.5$ | $267.4 \pm 15.0$ | $275.0 \pm 15.6$ |
| AT-DPT (sub. 20%) | $230.7 \pm 26.9$ | $233.7 \pm 25.5$ | $229.4 \pm 25.5$ | $231.7 \pm 27.2$ | $238.1 \pm 28.1$ | $241.9 \pm 32.0$ |
| AT-DPT (sub. 30%) | $201.5 \pm 24.5$ | $205.6 \pm 24.2$ | $195.9 \pm 21.0$ | $192.3 \pm 27.0$ | $202.3 \pm 23.5$ | $212.1 \pm 23.9$ |
| DPT | $229.6 \pm 7.1$ | $171.9 \pm 11.7$ | $215.0 \pm 8.1$ | $217.4 \pm 9.5$ | $273.8 \pm 7.8$ | $306.8 \pm 7.1$ |
| NPG$^\S$ | $244.1 \pm 7.5$ | $244.4 \pm 6.7$ | $239.5 \pm 9.4$ | $241.2 \pm 9.4$ | $248.9 \pm 8.0$ | $241.7 \pm 7.5$ |
| Q-learning$^\S$ | $240.3 \pm 5.4$ | $251.2 \pm 7.3$ | $236.8 \pm 5.8$ | $246.2 \pm 6.0$ | $244.7 \pm 6.8$ | $241.7 \pm 8.2$ |
| $\varepsilon = 0.4$ | | | | | | |
| AT-DPT | $242.2 \pm 11.9$ | $267.5 \pm 10.5$ | $241.7 \pm 10.2$ | $239.1 \pm 8.8$ | $258.2 \pm 11.8$ | $267.4 \pm 15.1$ |
| AT-DPT (sub. 10%) | $225.2 \pm 17.8$ | $245.5 \pm 17.9$ | $226.1 \pm 16.6$ | $222.7 \pm 17.6$ | $241.3 \pm 17.8$ | $250.2 \pm 18.8$ |
| AT-DPT (sub. 20%) | $199.4 \pm 20.4$ | $219.3 \pm 23.6$ | $200.5 \pm 22.2$ | $200.0 \pm 22.3$ | $208.5 \pm 25.8$ | $219.1 \pm 26.3$ |
| AT-DPT (sub. 30%) | $178.6 \pm 18.7$ | $195.5 \pm 23.9$ | $178.8 \pm 21.0$ | $176.2 \pm 20.7$ | $182.6 \pm 17.8$ | $190.5 \pm 17.6$ |
| DPT | $216.1 \pm 11.0$ | $143.5 \pm 11.0$ | $202.6 \pm 7.4$ | $205.9 \pm 7.8$ | $266.2 \pm 8.1$ | $306.8 \pm 7.1$ |
| NPG$^\S$ | $237.2 \pm 6.7$ | $243.7 \pm 7.9$ | $228.9 \pm 4.0$ | $228.1 \pm 8.1$ | $235.3 \pm 8.2$ | $241.7 \pm 7.5$ |
| Q-learning$^\S$ | $198.1 \pm 3.7$ | $238.6 \pm 6.0$ | $215.4 \pm 7.6$ | $224.7 \pm 7.3$ | $229.0 \pm 7.2$ | $225.6 \pm 5.4$ |

For the PPO baseline in Tables 5 and 11 we use the `cleanrl` implementation (Huang et al., 2022).

*Table 11.* Comparison of the average episode reward (higher is better) of the different algorithms under different attackers trained for 100 rounds in the **Miniworld environment**. Mean and 95% confidence interval (2×SEM) over 10 runs. Attack budget $B = 5$. $^{\S}$ PPO requires multiple episodes of online learning to converge to a stable policy; we run it for 100 episodes before evaluating the performance.

| Algorithm | Attacker Target | | Unif. Rand. Attack | Clean Env. |
| --- | --- | --- | --- | --- |
| | AT-DPT | DPT | | |
| | | $\varepsilon = 0.1$ | | |
| AT-DPT | $111.1 \pm 11.9$ | $114.1 \pm 13.0$ | $110.1 \pm 16.0$ | $123.9 \pm 16.7$ |
| DPT | $93.2 \pm 12.4$ | $92.8 \pm 14.2$ | $103.1 \pm 12.8$ | $110.0 \pm 14.7$ |
| PPO$^{\S}$ | $117.9 \pm 8.4$ | $115.5 \pm 4.1$ | $101.5 \pm 6.2$ | $123.5 \pm 8.1$ |
| | | $\varepsilon = 0.2$ | | |
| AT-DPT | $115.5 \pm 13.0$ | $114.0 \pm 17.5$ | $111.1 \pm 15.8$ | $114.9 \pm 20.2$ |
| DPT | $84.6 \pm 13.8$ | $90.0 \pm 14.8$ | $103.0 \pm 12.5$ | $110.0 \pm 14.7$ |
| PPO$^{\S}$ | $105.1 \pm 8.3$ | $109.9 \pm 9.0$ | $100.6 \pm 5.3$ | $123.5 \pm 8.1$ |
| | | $\varepsilon = 0.4$ | | |
| AT-DPT | $104.8 \pm 16.0$ | $116.8 \pm 18.8$ | $108.6 \pm 15.1$ | $112.7 \pm 23.9$ |
| DPT | $81.2 \pm 12.2$ | $70.2 \pm 15.0$ | $102.7 \pm 13.1$ | $110.0 \pm 14.7$ |
| PPO$^{\S}$ | $83.5 \pm 7.4$ | $83.8 \pm 7.2$ | $92.9 \pm 7.3$ | $123.5 \pm 8.1$ |

## B.6. Training curves

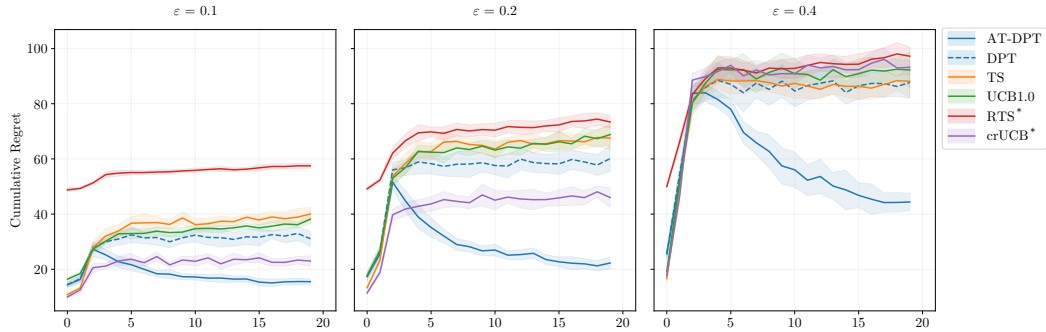

*Figure 4.* Adversarial training curves for training the attacker in the bandit setting, for different values of $\varepsilon$. Note, that in the case of AT-DPT it is trained along with the attackers. $^{*}$ We use tuned versions of RTS and crUCB, see Appendices A.1 and A.2 for more details.

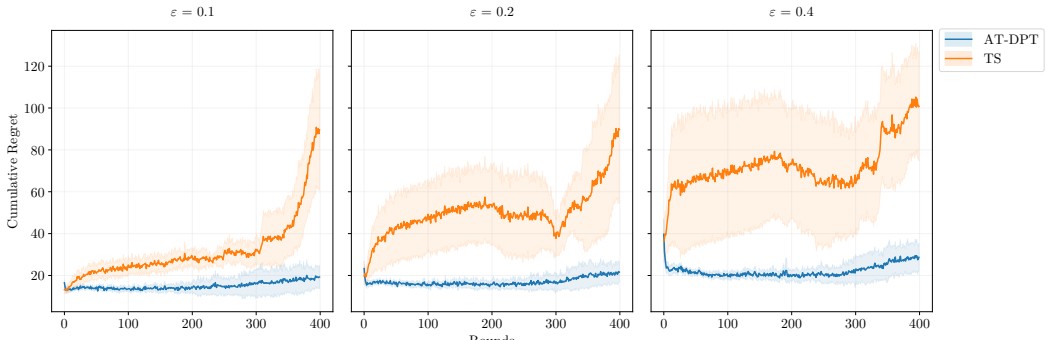

*Figure 5.* Adversarial training curves for training the adaptive attacker in the bandit setting, for different values of $\varepsilon$. Note, that in the case of AT-DPT it is trained along with the attackers.

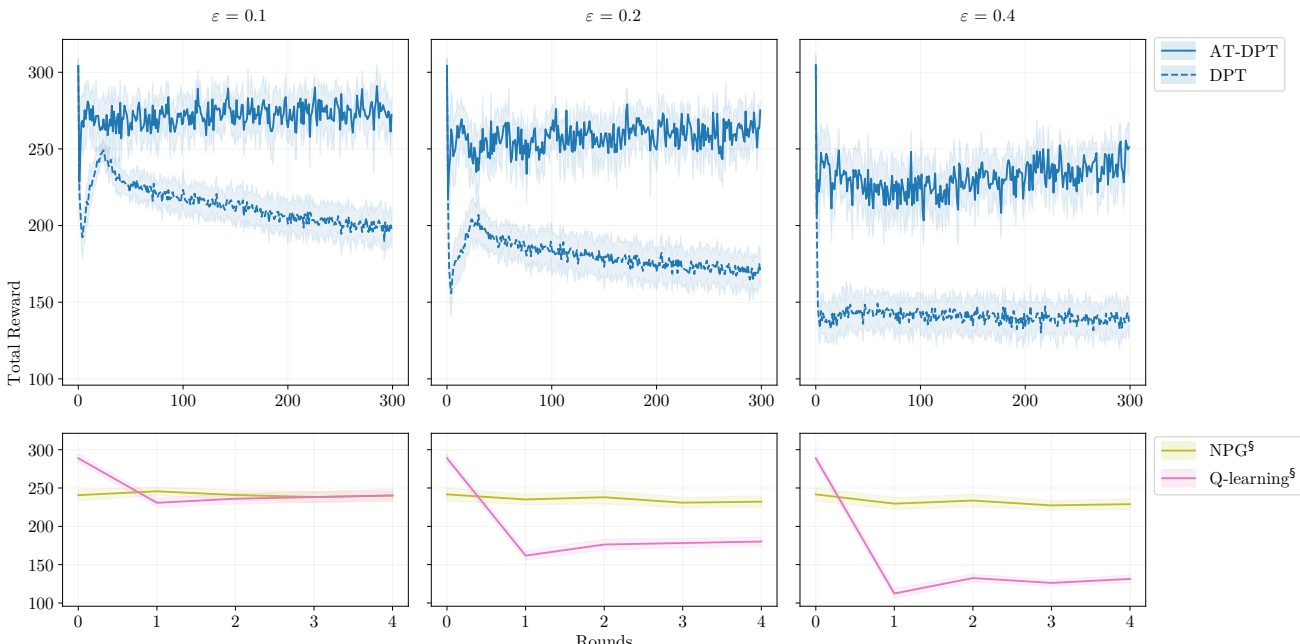

*Figure 6.* Adversarial training curves for training the attacker in the Darkroom2 environment, for different values of $\varepsilon$. Note, that in the case of AT-DPT it is trained along with the attackers. [§] NPG and Q-learning require multiple episodes of online learning to converge to a stable policy; we run them for 100 episodes before evaluating their performance.

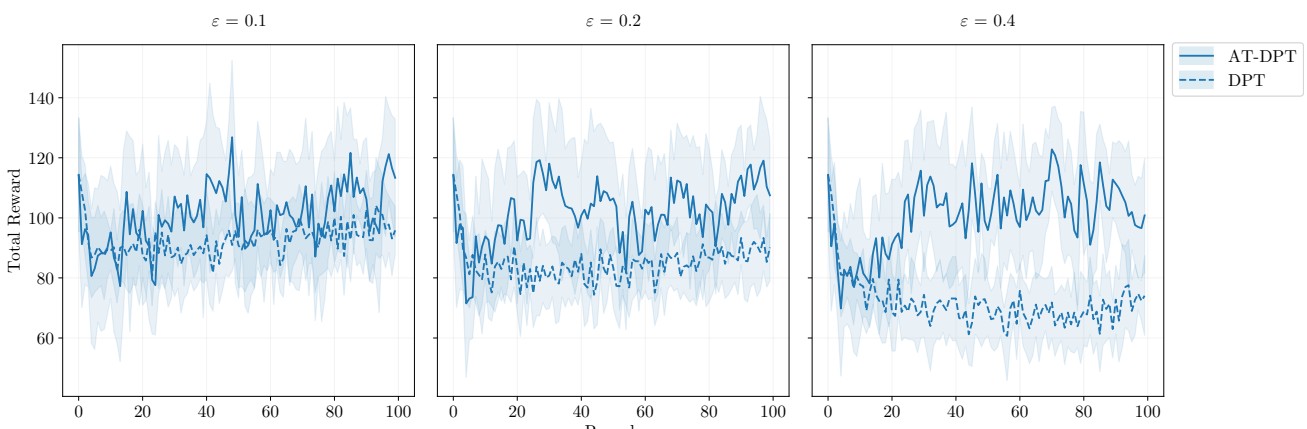

*Figure 7.* Adversarial training curves for training the attacker in the Miniworld environment, for different values of $\varepsilon$. Note, that in the case of AT-DPT it is trained along with the attackers.

# C. Further details

## C.1. Compute resources

The experiments were run on a compute cluster with NVIDIA A100 80GB PCIe and NVIDIA H100 94GB NVL GPUs. Approximate GPU machine runtime of experiments, per run:

- Bandit Environment:
  - Pretraining – 3.4 h
  - Adversarial training – 0.4 h
  - Evaluation – 0.6 h

- Bandit Environment, Adaptive Attacker:
  - Pretraining – 3.4 h (same as bandit env.)
  - Adversarial training – 0.6 h
  - Evaluation – 0.2 h

- Darkroom2 Environment:
  - Pretraining – 1.4 h
  - Adversarial training – 0.6 h
  - Evaluation – 3.4 h[§]

- Miniworld Environment:
  - Pretraining – 13.1 h
  - Adversarial training – 2.7 h
  - Evaluation – 0.9 h

[§] NPG and Q-learning required multiple episodes of online learning before converging to a stable policy, therefore leading to an increased evaluation run time.

## C.2. Interpretation of attack in Darkroom2

We present an illustration of an example environment and attacker's strategy in Figure 8, taken from the middle of a sample AT-DPT adversarial training run. The attacker's strategy observed in the illustration shows the attack is not arbitrary – it is focusing on states nearby the goal. We can see an attack of $+1$ on a goal which gives a reward of 2 – this would change the observed reward into 3. A reward value of 3 was not seen during pretraining DPT, and in this round, upon encountering this it provokes undesirable behavior (*stay* at a low-reward state), causing a low episode reward. During the next round of training we find that the DPT has learned to recover from this mistake, and given the same attacker's strategy for that state successfully ignores this attack.

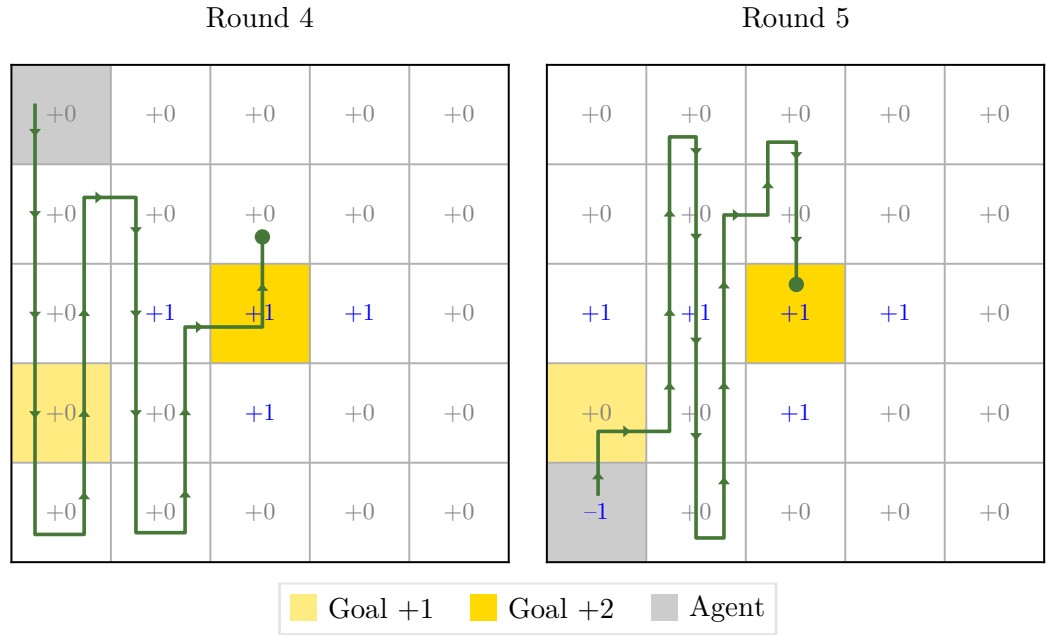

Figure 8. An illustration of the Darkroom2 environment with an attacker's poisoning strategy during a sample training run. Gray and blue numbers $-1$, $+0$, and $+1$ indicate the attacker's current poisoning strategy. Green path denotes trajectory taken by the agent for that round; green circle indicates the state where the agent chose to stop and exploit the current reward by choosing the stay action.

## C.3. AT-DPT test phase

---

**Algorithm 3** AT-DPT test phase

---

1: **input:** victim $\pi_\theta$ – AT-DPT with params $\theta$
2: **input:** attacker $\pi_\phi^\dagger$ with params $\phi$, budget $B$, fraction of steps poisoned $\varepsilon$
3: Sample $M$ tasks $\{\mathcal{M}_i \sim \mathcal{T}\}_{i=1}^m$
4: **for** all $\mathcal{M}_i$ simultaneously **do**
5:     $s_0 \sim \rho_{\mathcal{M}_i}$
6:     $D^\dagger \leftarrow \{\}$
7:     **for** $h = 0, \ldots, H-1$ **do**
8:         select action $a_h \sim \pi_{\theta_n}(\cdot \mid D^\dagger, s_h)$
9:         $\tilde{r}_h = \begin{cases} r_h^\dagger \sim \pi_\phi^\dagger(\cdot \mid s_h, a_h, \bar{r}_h) & \text{with probability } \varepsilon \\ \bar{r}_h \sim R(\cdot \mid s_h, a_h) & \text{otherwise} \end{cases}$
10:         $s_{h+1} \sim T(\cdot \mid s_h, a_h)$
11:         append $(s_h, a_h, \tilde{r}_h, s_{h+1})$ to $D^\dagger$
12:     **end for**
13: **end for**

---

