# OpenReview forum: "Robust In-Context Reinforcement Learning Under Reward Poisoning Attacks"
_ICML.cc/2026/Conference — ICML 2026 regular_

### Official Review · Reviewer_KyRn · 2026-03-11

**Soundness:** 2
**Presentation:** 3
**Significance:** 3
**Originality:** 2
**Overall Recommendation:** 4
**Confidence:** 3

**Summary:**

This paper studies test-time reward poisoning attacks against in-context RL models, focusing on the Decision-Pretrained Transformer (DPT). The main idea is that, unlike standard RL policies whose test-time behavior is independent of rewards, an in-context RL agent updates its policy  through the history of data, so corrupting rewards at test time can alter the learning algorithm implemented in-context. The paper proposes AT-DPT, an adversarially trained variant of DPT.  The method is evaluated in multi-armed bandits, linear bandits, and simple  MDPs with both adaptive and non-adaptive attackers. Empirically, the paper reports that AT-DPT is substantially more robust than vanilla DPT.

**Compliance With Llm Reviewing Policy:**

Affirmed.

**Final Justification:**

I thank the authors for their rebuttal and follow-up clarifications, especially in light of some of my original scores.  I agree that the underlying problem is important, because in ICRL reward poisoning is a genuinely test-time vulnerability since rewards enter the context used for online adaptation.

- In general the empirical results are interesting, and the paper is written well, and this is why I view the paper as having clear merits. In light of this, I updated my significance/originality/quality/clarity.

- I have still concerns about methodological incrementality and missing theory remain, but the discussion changed my view on whether the current evidence is already sufficient for a borderline accept.   After re-reading the paper and the overall discussion, what changed  my mind most is that the empirical signal is stronger than I initially credited, and the added Miniworld results were also helpful on scope although these were added latter. While they do not resolve  the issue about architectural breadth, they do provide nontrivial evidence beyond the smallest environments.

- The discussion also partly addressed my attacker-family-mismatch concern, and the rebuttal/follow-up clarified results.
- That said, my key reservations still stand in substance: AT-DPT remains a natural adversarial-training extension of DPT and the evidence is still DPT-centric. Formal results are left for  analysis to future work.

Overall I believe the paper only requires marginal modifications to  allow other people  to build on top of it, and with the proposed new results/changes I now support a weak accept.

**Key Questions For Authors:**

1. How robust is AT-DPT to attacker-family mismatch? The current results suggest some cross-attacker generalization, but the attackers still appear structurally close to those used during training.
2. How dependent is AT-DPT on access to clean rewards and oracle optimal actions during training? this is a strong assumption

**Limitations:**

Yes

**Strengths And Weaknesses:**

** Strenghts**
1. The paper is original in its problem formulation (to my knowledge). In in-context RL reward poisoning can be a test-time attack because rewards are part of the context through which the agent adapts online.
2. The problem is important. Security and robustness for in-context decision-making systems are likely to matter more, not less, as sequence-modeling approaches become more common.
3. The paper is generally readable and the central narrative is easy to follow. The motivation for why test-time reward poisoning matters specifically for ICRL is clear.


** Weaknesses**
1. Methodologically, AT-DPT is a natural adversarial-training construction built on DPT, and it combines established ingredients. So the method is more a novel application and synthesis than a fundamentally new algorithm. While the problem itself is novel, the solution is somewhat incremental.
2.  The paper is almost entirely empirical, without any theoretical claims on robustness guarantee.
3. The experiments are still narrow for the strength of the claims. The environments are all relatively small-scale, and the authors mainly focus on DPT, which is  a transformed-based architecture . However, in-context RL could also be implemented using LSTMs, and DPT is not necessarily the only method to train in-context RL policies (see for example RL^2, Vintix or Russo et al 2025, In-Context Learning for Pure Exploration). Overall, I believe the results are too narrow for the claims.
4. Third, the attack model is strong but also somewhat specialized. The attacker has access to the underlying clean reward when producing poisoned rewards, and training requires access to clean environments and oracle optimal actions. That is acceptable as a first threat model, but it reduces practical realism.

---

> ### Author Rebuttal · Authors · 2026-03-30
>
> Thank you for the review and the set of constructive comments and questions. We are glad to see that the reviewer found the problem original and important (strengths 1 & 2), but we are confused about the given significance and originality score. We respond to the weaknesses and questions below.
>
> W1. It is a standard practice that we use established algorithms as a basis for new methods, and we believe that building on top of a standard algorithm allows us to solve it with appropriate changes; we do not see this point as a weakness of our work.
>
> W2. While we do not provide theoretical results, we believe our framework is well-motivated and grounded in standard principles of adversarial training. It is formulated as a game between an attacker and a defender; solving this game directly implies robustness. Therefore, it is not surprising that we do not offer formal guarantees – finding a Nash equilibrium is, in general, a hard problem and out of the scope of this paper. Moreover, in the robust RL literature, it is common to focus on empirical evaluation, especially for techniques based on adversarial training. For example, consider the following works:
>
> Pinto et al., Robust Adversarial Reinforcement Learning, ICML 2017
>
> Pattanaik et al., Robust Deep Reinforcement Learning with Adversarial Attacks, AAMAS 2018
>
> Lin et al., Model-based Adversarial Meta-Reinforcement Learning, NeurIPS 2020
>
> Oikarinen et al., Robust Deep Reinforcement Learning through Adversarial Loss, NeurIPS 2021
>
> Note that some of these papers specifically consider adversarial attacks. We believe our work can spark valuable discussion and inspire future research with a theoretical focus.
>
> W3. We provide an additional set of results in a high-dimensional environment Miniworld (continuous state space, 3D observation), please see Appendix B.4, Table 9. These results show that, similarly to other environments, AT-DPT can be robust to adversarial reward corruption. That is, there is evidence to show that the method scales to larger environments.
>
> In this work we build on top of DPT, and similar experiments can be run on different methods, but we believe it is out of scope for this paper. Some methods, such as AD [1] may not fit our setting, see a brief discussion on AD in Appendix A in the paper. We appreciate the additional references and will add them to the final version of the manuscript. Our goal was to demonstrate the main principle of adversarial training for robust ICRL, and to our knowledge, this type of meta-learning problem has not been considered in previous work on reward poisoning attacks, yet it offers certain advantages over common approaches. Notably, it enables the learning of algorithms that are robust to a specific task distribution of interest, potentially leading to better performance and generalization.
>
> W4. We do not think that the attack is too specialized – we believe that the attacker having access to the underlying clean reward makes it stronger, and we do not see it as a weakness, but rather a strength. In addition, we can see for example in Table 1 these attacks work well for general robust versions of algorithms (e.g., crUCB, RTS), therefore we have to disagree that it is specialized. Please also see an ablation replacing a fraction of expert actions with suboptimal ones in our response to reviewer Him6 – we find that strict reliance on optimal actions is not necessary. With regard to realism, some tasks may be easy to solve in an uncorrupted environment, but more difficult with reward poisoning. Simulating such an environment with an adversary would make practical sense to increase robustness. This kind of setup is not too far from red-teaming with LLMs – we inject “noised” prompts to elicit a “bad reward”, and later fine-tune the model to be robust against these prompts.
>
> Q1. Our results indeed show cross-attacker generalization, but other attacker families have not been tested. We believe our attacks are sufficiently strong and general enough because they are able to fool algorithms which have been specifically designed to handle corruption robustness . We would like to disagree with the reviewer that the attackers are structurally close to those during training. Take for example columns in Table 4 – the attack that was learnt for DPT and the one learnt for NPG give significantly different outcomes for the different algorithms. Perhaps there is a specific attacker you would like to see?
>
> Q2. We performed additional experiments studying the effect of replacing a fraction of expert actions with suboptimal ones – please see our response to reviewer Him6. We find that reliance on the expert is not critical, and there is evidence that a robust policy can be learned from slightly weaker demonstrations.
>
> We would be grateful if you could let us know if there are still questions or outstanding issues you have with the paper.
>
> [1] Laskin et al., In-context Reinforcement Learning with Algorithm Distillation, ICLR 2023.

---

> > ### Author Rebuttal · Reviewer_KyRn · 2026-04-03
> >
> > Thank you for the rebuttal.  The rebuttal partially improves my view.
> >
> >
> > - On experimental scope
> >   - My main concern remains. The ICRL method is always DPT, while the claims are framed more broadly.
> >   - The Miniworld result helps, but evidence is still quite limited in my view and  it does not really address architectural breadth. So I would describe the empirical evidence as promising.
> >
> > - On novelty
> >   - My view is unchanged that the problem formulation is the most novel part of the paper. Test-time reward poisoning is indeed especially relevant for ICRL because rewards enter the context. My concern was about the *method*, not about whether it is acceptable to build on standard components. AT-DPT still reads as a natural adversarial-training extension of DPT, rather than a fundamentally new algorithm. I do not think this makes the paper uninteresting. But I do think it matters when evaluating methodological originality.
> >
> > - On theory
> >   - I understand the authors' point, however, this does not really remove the concern. The paper remains almost entirely empirical, and the discussion section explicitly leaves an analysis to future work. Prior work, e.g. Zhang et al., “Adaptive Reward-Poisoning Attacks against Reinforcement Learning” (ICML 2020)  or Nika et al., “Online Defense Strategies for Reinforcement Learning Against Adaptive Reward Poisoning” (AISTATS 2023) provide also formal results (or the DPT paper itself).
> >   - Since the defense is formulated as a game between attacker and defender, I still view the lack of any formal analysis as a weakness.
> >
> > - On the attack model and training assumptions
> >   - My concern was not that this threat model is invalid, but that it is somewhat specialized and may be less realistic for some applications.
> >   - Regarding the  weaker demonstrations, this is potentially interesting, but since those results are not in the paper or in this rebuttal, I cannot weigh them heavily.
> >   - On attacker-family mismatch: The rebuttal partly addresses this concern,  however, the corruption mechanism remains within the same overall famil to my understanding.
> >
> >
> >
> >
> > To be clear, I do think the bandit and linear-bandit results are interesting.  This is why I view the paper as having clear merits despite my concerns. While I appreciate the clarifications and would soften some of my original wording, my overall recommendation would remain unchanged.

---

> > > ### Author Response · Authors · 2026-04-05
> > >
> > > Thank you for the follow-up response. We respectfully disagree that the remaining concerns merit rejection. We address each point below.
> > >
> > > **Experimental scope**: We are a bit surprised that the reviewer is under the impression that our claims are framed more broadly, since our contributions clearly state that we focus on one method. We would like to clarify that we do not claim that we provide a training framework applicable to all existing ICRL algorithms, but that we can design an ICRL algorithm robust to corruption and that the DPT approach we introduce can yield significant performance gains. There is also a good reason for focusing on DPT, and not, for example, AD or any of its extensions. AD distills a _learning algorithm's improvement trajectory_, meaning that it tries to imitate a learning algorithm. In our case, AD would need a source RL algorithm that already knows how to improve under reward poisoning – the exact problem we are trying to solve. We provide a new set of experiments showcasing AD as a baseline – please see the Reply Rebuttal Comment to Reviewer bk5T.
> > >
> > > We are also surprised that the reviewer believes that our experimental testbed does not address architectural breadth, as we follow the experimental design introduced in the DPT paper, published at NeurIPS.
> > >
> > > **Novelty**: At a high-level our approach is a combination of existing techniques (DPT and adversarial training), but this does not mean it's not worth exploring nor that our results are trivial, especially because we are exploring this combination in a novel context. We would argue that studying existing algorithms under different contexts is quite common in the literature, and in fact, one of the references that the reviewer provided (Nika et al., AISTATS 2023) does that as well: it takes existing online learning algorithms (Exp3, OMDUCB), and utilizes them to optimize hyper-parameters of an existing defense to reward poisoning approach. Regardless, the mentioned work is valuable because it applied known methods to a novel and important problem setting – which is, we believe, exactly what we do, but in the previously unstudied meta-RL/ICRL setting.
> > >
> > > **Theory**: We believe that the comparison to the works the reviewer provided may not be directly applicable, as they consider different and more simplified formal settings:
> > > - Zhang et al. (ICML 2020) focused on studying poisoning attacks on a victim (learning algorithm) oblivious to its presence, with the goal of maximizing the performance of poisoning. In our case, the goal is to design a robust victim, which is much more challenging, and this is well-recognized in the community.
> > > - Nika et al. (AISTATS 2023) considers a fixed attacker model and a fixed victim model – these models are given, they are not optimized to be in an equilibrium – and optimizes their hyperparameters (each assumed to be from a finite set) via no-regret learning algorithm. These hyperparameters have to come from a small set in order to make the approach practical, as evident from their experiments. Furthermore, in their experiments the attack is not optimized for the defense model. In our case we optimize attack and victim models which, given we consider neural networks, have a much larger parameter space.
> > > - The DPT setting doesn't consider corruption, and this makes our setting much more challenging as it changes the nature of Assumption 1. While we can draw theoretical insights by adapting their analysis to our setting (which is what our discussion in Section 6 is based on), this adaptation requires that AT-DPT converges to an equilibrium of the underlying game. Hence, an analogous assumption to Assumption 1 for our setting is to assume such convergence. However, we believe that this may be too strong to assume, as convergence to an equilibrium is often non-trivial to show.
> > >
> > > Finally, we would like to point out that many works on adversarial training are empirical in nature, and we explicitly acknowledge that a substantial theoretical contribution could be potential future work.
> > >
> > > **Attacker-family mismatch**: Regarding the experiments with suboptimal expert demonstrations, we plan to run these experiments for other environments and other values of $\varepsilon$ and include them in the final version of the paper. We would also like to mention the additional experiments provided in our response to Reviewer 1E2G, where we show that AT-DPT trained with a weak attacker (B=1) transfers robustness to substantially stronger attacks (B=3 and B=5). In particular, AT-DPT trained at B=1 and evaluated at B=3 achieves regret of approx. 28-34 across all attacker targets, compared to 82-107 for the baselines. This shows more evidence that the learned robustness generalizes beyond the specific attack family and magnitude seen during training, hopefully addressing your concern about attacker-family mismatch.
> > >
> > > We hope that you will reconsider your assessment in light of these clarifications and the additional results provided.

---

### Official Review · Reviewer_1E2G · 2026-03-11

**Soundness:** 2
**Presentation:** 3
**Significance:** 2
**Originality:** 2
**Overall Recommendation:** 4
**Confidence:** 4

**Summary:**

The paper introduces an adversarial training framework, AT-DPT, designed to improve the corruption-robustness of in-context reinforcement learning models against test-time reward poisoning attacks. The method simultaneously optimizes a population of reward-poisoning adversaries—constrained by a soft budget—and a Decision-Pretrained Transformer (DPT) that learns to predict optimal actions from the corrupted contexts. Empirical evaluations span multi-armed bandits, linear bandits, and gridworld/visual MDPs, comparing the proposed approach against standard and robust baseline algorithms.

**Compliance With Llm Reviewing Policy:**

Affirmed.

**Final Justification:**

I have read the rebuttal, and I am willing to raise my score

**Key Questions For Authors:**

-- Can you provide empirical evidence or ablations showing how AT-DPT performs when the oracle optimal actions are replaced with sub-optimal or noisy expert demonstrations during pretraining?

-- What happens to the performance of AT-DPT if the test-time attacker strictly violates the training-time budget $B$? How brittle is the learned policy to out-of-distribution attack magnitudes?

**Limitations:**

The adversarial robustness demonstrated is constrained strictly to the $\epsilon$-contamination model and the predefined bounds explored during training. It is highly unlikely this empirical robustness generalizes to entirely different attack vectors, such as observation poisoning or unbounded reward manipulations.

Another limitation revolves around the fundamental reliance on optimal action labels for the DPT architecture.

other points can follow weaknesses above

**Strengths And Weaknesses:**

Strengths:


-- In general, the paper is well written and easy to follow.

-- Extends the study of reward poisoning from traditional single-task RL to the meta-RL/in-context learning paradigm, addressing a valid vulnerability in models like DPT.

-- The empirical evaluation spans standard and adaptive attackers across bandit and MDP environments.

--- Cross-validation evaluation against attackers trained on different target algorithms provides a realistic assessment of out-of-distribution adversarial generalization.

Weakness:

-- The paper lacks formal convergence guarantees for the proposed adversarial training procedure. While the authors present a bi-level optimization objective aiming for a Nash equilibrium, they do not theoretically establish that AT-DPT actually converges to an approximate equilibrium. The theoretical justification relies heavily on loose analogies to posterior sampling rather than rigorous bounds.

-- The training protocol relies entirely on an oracle providing optimal actions $a^{*}$ during the supervised learning phase. This assumption is highly restrictive for general RL settings where the optimal policy is unknown, severely limiting the practical applicability of the method to environments where the task is already solved.

---

> ### Author Rebuttal · Authors · 2026-03-30
>
> Thank you for your valuable comments and feedback. We are pleased to see that your view on our work is overall positive.
>
> W1. While we do not provide theoretical results, we believe our framework is well-motivated and grounded in standard principles of adversarial training. It is formulated as a game between an attacker and a defender; solving this game directly implies robustness. Therefore, it is not surprising that we do not offer formal guarantees – finding a Nash equilibrium is, in general, a hard problem.
>
> Moreover, in the robust RL literature, it is common to focus on empirical evaluation, especially for techniques based on adversarial training. For example, consider the following works:
>
> Pinto et al., Robust Adversarial Reinforcement Learning, ICML 2017
>
> Pattanaik et al., Robust Deep Reinforcement Learning with Adversarial Attacks, AAMAS 2018
>
> Lin et al., Model-based Adversarial Meta-Reinforcement Learning, NeurIPS 2020
>
> Oikarinen et al., Robust Deep Reinforcement Learning through Adversarial Loss, NeurIPS 2021
>
> Note that some of these papers specifically consider adversarial attacks. We believe our work can inspire future research with a theoretical focus.
>
> To further support this argument, we would like to emphasize that our approach is conceptually different from those in prior work. Specifically, we aim to provide a principled framework for learning practical RL algorithms that are robust to corruption. To our knowledge, this type of meta-learning problem has not been considered in previous work on reward poisoning attacks, yet it offers certain advantages over common approaches. Notably, it enables the learning of algorithms that are robust to a specific task distribution of interest, potentially leading to better performance and generalization.
>
> Q1. We provide additional experiments showing training with suboptimal actions, please see the response to reviewer Him6. We find that increasing the percentage of suboptimal actions slightly degrades performance, but it is not too substantial.
>
> Q2. We test on out-of-distribution attacks in all experiments, meaning the test-phase attacks differ from the train-phase attacks. In addition to this, we provide a set of additional experiments showing how training on a weaker attacker (B=1) transfers to stronger attacks (B=3, B=5). The setup is otherwise the same as in Table 1 ($\varepsilon = 0.4$).
>
> **Train B=1, eval B=1**
> ||AT-DPT|DPT|TS|RTS|UCB|crUCB|Unif. Rand.|Clean|
> |-|-|-|-|-|-|-|-|-|
> |AT-DPT|29.3 ± 2.0|28.2 ± 2.2|28.4 ± 1.5|28.6 ± 1.8|28.5 ± 2.1|28.5 ± 2.5|31.7 ± 2.4|14.7 ± 1.9|
> |DPT|35.0 ± 2.2|31.6 ± 1.7|33.3 ± 2.1|32.8 ± 1.6|32.7 ± 2.0|33.0 ± 3.1|28.2 ± 1.9|11.5 ± 0.5|
> |TS|34.3 ± 2.0|29.9 ± 1.2|30.1 ± 1.0|31.8 ± 1.3|30.3 ± 1.7|30.8 ± 2.3|26.7 ± 2.2|8.7 ± 0.6|
> |RTS|34.3 ± 1.9|30.9 ± 1.4|30.6 ± 1.8|31.2 ± 1.6|30.2 ± 0.9|30.7 ± 1.6|27.1 ± 1.6|10.2 ± 0.4|
> |UCB|40.1 ± 1.6|36.2 ± 1.4|37.2 ± 1.2|37.2 ± 1.8|36.6 ± 1.7|36.2 ± 1.5|31.4 ± 1.5|16.0 ± 0.5|
> |crUCB|37.1 ± 1.4|33.6 ± 1.5|34.3 ± 1.5|33.1 ± 1.6|33.3 ± 1.8|34.6 ± 1.9|26.9 ± 0.9|15.8 ± 0.5|
>
> **Train B=1, eval B=3**
> ||AT-DPT|DPT|TS|RTS|UCB|crUCB|Unif. Rand.|Clean|
> |-|-|-|-|-|-|-|-|-|
> |AT-DPT|28.5 ± 1.6|29.3 ± 1.4|34.1 ± 2.9|31.2 ± 2.4|28.9 ± 2.1|27.8 ± 1.7|40.3 ± 3.1|14.7 ± 1.9|
> |DPT|63.5 ± 6.6|58.2 ± 5.4|61.4 ± 7.5|59.8 ± 7.7|57.4 ± 8.1|58.0 ± 6.5|35.6 ± 3.0|11.5 ± 0.5|
> |TS|107.1 ± 2.4|97.8 ± 4.1|94.1 ± 3.8|94.9 ± 5.6|90.8 ± 2.4|92.4 ± 4.5|34.4 ± 3.1|8.7 ± 0.6|
> |RTS|105.8 ± 2.4|97.3 ± 4.3|90.4 ± 4.4|93.0 ± 4.6|89.9 ± 3.9|87.9 ± 2.7|33.7 ± 2.0|10.2 ± 0.4|
> |UCB|105.6 ± 3.0|95.3 ± 4.8|91.1 ± 4.2|90.3 ± 4.9|88.7 ± 3.2|91.4 ± 3.4|39.2 ± 2.1|16.0 ± 0.5|
> |crUCB|85.7 ± 3.7|84.3 ± 2.2|81.9 ± 4.4|82.1 ± 3.7|79.3 ± 2.9|81.9 ± 4.9|31.8 ± 1.4|15.8 ± 0.5|
>
> **Train B=1, eval B=5**
> ||AT-DPT|DPT|TS|RTS|UCB|crUCB|Unif. Rand.|Clean|
> |-|-|-|-|-|-|-|-|-|
> |AT-DPT|34.9 ± 1.7|35.7 ± 2.3|40.5 ± 3.5|37.7 ± 2.6|35.5 ± 2.1|35.0 ± 2.9|69.8 ± 12.6|14.7 ± 1.9|
> |DPT|58.7 ± 8.4|56.1 ± 11.2|61.7 ± 9.3|59.2 ± 10.0|57.1 ± 9.7|56.6 ± 10.6|58.3 ± 10.0|11.5 ± 0.5|
> |TS|65.0 ± 5.5|59.5 ± 6.1|65.3 ± 4.1|60.9 ± 6.1|53.7 ± 3.8|59.8 ± 4.3|54.9 ± 10.4|8.7 ± 0.6|
> |RTS|61.5 ± 5.3|57.9 ± 4.9|62.3 ± 5.3|61.2 ± 6.1|53.1 ± 4.7|56.8 ± 3.7|53.4 ± 10.6|10.2 ± 0.4|
> |UCB|60.1 ± 4.6|53.2 ± 4.5|64.4 ± 4.4|58.5 ± 5.5|50.5 ± 3.9|52.2 ± 3.9|59.5 ± 12.1|16.0 ± 0.5|
> |crUCB|79.9 ± 2.9|80.2 ± 3.6|82.0 ± 4.4|81.6 ± 3.2|75.3 ± 5.1|80.0 ± 3.4|41.9 ± 6.5|15.8 ± 0.5|
>
> These results seem to show that there is evidence that AT-DPT transfers robustness to stronger attacks. That is, AT-DPT trained with a weaker type of attack learns to be robust against a stronger attacker.
>
> We think it would be interesting to consider generalization to different attack vectors, such as observation or action poisoning, in future work, and there is potential to apply an approach similar to ours.
>
> We hope we have addressed your main questions and clarified any important points. We would be grateful if you could let us know of any other outstanding issues or questions you have with the paper.

---

> > ### Author Rebuttal · Reviewer_1E2G · 2026-04-03
> >
> > Thanks for the rebuttal. Given my questions are well addressed, I will raise my score.

---

> > > ### Author Response · Authors · 2026-04-05
> > >
> > > Thank you very much for acknowledging our previous response and for raising your score. We greatly appreciate your effort to understand and improve the paper.

---

### Official Review · Reviewer_Him6 · 2026-03-12

**Soundness:** 3
**Presentation:** 3
**Significance:** 3
**Originality:** 3
**Overall Recommendation:** 5
**Confidence:** 3

**Summary:**

The paper focuses on In Context Reinforcement Learning of adversarial attacks on DPTs.The attackers minimize the agent's true return under a soft budget constraint, while the agent learns to infer optimal actions from the poisoned context.

**Compliance With Llm Reviewing Policy:**

Affirmed.

**Final Justification:**

My comments were addressed in the rebuttals, and thus I maintain my initial positive scores. I think the work address an interesting problem, and while using an oracle is a limitation, it still makes meaningful contributions.

**Key Questions For Authors:**

1. Typically when training multiple models simultaneously, it is common to observe mode collapse or training instabilities. Did the authors observe any such downsides during training?

**Strengths And Weaknesses:**

Strengths:
1. Clean bi-level formulation for ICRL attacks
2. Generalization across attacker algorithms: AT-DPT trained against one attacker generalizes well to attackers targeting other algorithms

Weaknesses:
1. Oracle requirement is a significant limitation. The method requires access to optimal actions during adversarial training.

---

> ### Author Rebuttal · Authors · 2026-03-30
>
> Thank you for the review and feedback. We are pleased to see that your view on our work is positive.
>
> W1. We provide additional experiments showcasing the effect of having suboptimal expert actions. The experiment setup is the same as in Table 1 ($\varepsilon = 0.4$). Rows indicated by _subopt._ mean that this percentage of timesteps in an episode is chosen at random to be labelled suboptimally by the expert:
>
> ||AT-DPT|DPT|TS|RTS|UCB|crUCB|Unif. Rand.|Clean|
> |-|-|-|-|-|-|-|-|-|
> |AT-DPT|23.7 ± 1.8|24.1 ± 2.0|29.5 ± 3.2|26.9 ± 1.8|24.3 ± 1.1|23.4 ± 1.7|37.7 ± 3.1|13.2 ± 0.7|
> |AT-DPT (subopt. 10%)|30.6 ± 2.3|30.9 ± 2.8|35.9 ± 3.7|32.8 ± 2.2|30.9 ± 2.4|30.3 ± 2.5|41.6 ± 2.4|17.4 ± 1.8|
> |AT-DPT (subopt. 20%)|35.4 ± 2.4|35.8 ± 3.2|40.0 ± 3.9|37.9 ± 2.4|36.0 ± 2.7|34.7 ± 2.9|42.9 ± 2.9|21.0 ± 2.9|
> |AT-DPT (subopt. 30%)|41.2 ± 2.9|41.9 ± 3.8|45.7 ± 3.3|43.6 ± 2.5|41.6 ± 3.0|41.0 ± 3.4|47.8 ± 3.5|25.9 ± 3.3|
> |DPT|65.1 ± 8.0|59.7 ± 5.1|62.0 ± 8.5|60.2 ± 7.2|55.3 ± 7.8|58.6 ± 7.7|35.7 ± 3.7|11.5 ± 0.5|
>
> We find that increasing the percentage of suboptimal actions slightly degrades performance, but it is not too substantial, and there is evidence that a robust policy can be learned from slightly weaker demonstrations.
>
> Q1. We provide a set of learning curves in Appendix B.5 and describe our observations regarding stability as follows. In AT-DPT we see an initial few rounds of low return (or high regret), followed by an upward trend for the return (or downward trend for the regret). In addition, we hypothesize that with a bounded budget instabilities are not likely, because eventually the budget will be exhausted and training converges.
>
> We hope we have answered your questions and once again thank you for the feedback.

---

> > ### Author Rebuttal · Reviewer_Him6 · 2026-04-02
> >
> > My concerns have been adequately addressed. I will keep my positive rating.

---

> > > ### Author Response · Authors · 2026-04-05
> > >
> > > Thank you for the acknowledgement, positive feedback, and time spent reviewing our paper.

---

### Official Review · Reviewer_bk5T · 2026-03-13

**Soundness:** 3
**Presentation:** 3
**Significance:** 2
**Originality:** 4
**Overall Recommendation:** 4
**Confidence:** 4

**Summary:**

This paper studies the robustness of in-context reinforcement learning (ICRL) to test-time reward poisoning attacks, focusing on the Decision-Pretrained Transformer (DPT).
The authors propose Adversarially Trained DPT (AT-DPT), a framework that simultaneously trains a population of reward-poisoning attackers (via REINFORCE) and a DPT defender (via supervised learning on corrupted contexts) in a min-max fashion.
Experiments span multi-armed bandits, linear bandits, and a gridworld MDP (Darkroom2), showing that AT-DPT outperforms classical robust bandit algorithms (RTS, crUCB) and vanilla DPT under learned attackers, and generalizes to unseen attack types.
The paper also provides informal theoretical insight into the connection between AT-DPT and corrected posterior sampling under Huber's epsilon-contamination model.

**Compliance With Llm Reviewing Policy:**

Affirmed.

**Final Justification:**

My concerns have been fully addressed, and I have decided to maintain my score.

**Key Questions For Authors:**

1. How does AT-DPT perform when the attacker uses a substantially more powerful architecture (e.g., larger transformer, PPO optimization, or full context access)? The current attacker capacity is not ablated—please provide experiments that vary the attacker size and the optimization method.
2. What happens at higher contamination levels (e.g., epsilon = 0.6, 0.8)? Is there a phase transition where AT-DPT's advantage disappears?
3. Why is ARDT (Tang et al., NeurIPS 2024) not included as an experimental baseline, and does AT-DPT's adversarial training framework transfer to at least one non-DPT ICRL backbone (e.g., Algorithm Distillation)?

**Limitations:**

The authors discuss limitations honestly in Section 6, including the requirement for an oracle action, the clean-environment trade-off, and the single attack specification per experiment.
However, the scalability limitation is underemphasized -- all environments are toy-scale, and it is unclear whether the adversarial training framework can scale to environments with continuous state/action spaces, high-dimensional observations, or long horizons.

**Strengths And Weaknesses:**

**Strengths:**

- **S1. Novel and well-motivated problem formulation.**
Studying test-time reward poisoning in ICRL is novel.
Prior corruption-robust RL work focuses on training-time attacks, but ICRL's in-context nature means rewards at test time directly affect the learned policy.

- **S2. Thorough experimental evaluation with strong baselines.**
The paper compares against a broad set of baselines, including theoretically-motivated robust algorithms (RTS, crUCB, CRLinUCB) and extensive ablations.

- **S3. Cross-attack generalization.**
Table 1 demonstrates that AT-DPT trained against its own attacker also performs well when evaluated against attackers trained to target other algorithms (TS, RTS, UCB, crUCB).
This is a practically important property, since a defender cannot know the attacker's strategy in advance.

**Weaknesses:**

- **W1. Toy-scale environments with no evidence of scalability.**
All experiments use trivially small environments: 5-armed bandits, 10-armed linear bandits with d=2, and a 5x5 gridworld with 25 states and 5 actions.
The paper's motivation (Section 6) invokes LLM-based agents and RAG poisoning, yet the gap between this motivation and the experimental scale is enormous, and adversarial training already incurs substantial compute at this toy scale.
The paper does not discuss how the simultaneous training of M separate attacker networks scales to larger environments, leaving it unclear whether the approach is feasible beyond the settings tested.

- **W2. Practical robustness concerns: clean-environment degradation.**
AT-DPT consistently underperforms vanilla DPT in clean environments (e.g., Table 1: AT-DPT 13.0 vs. DPT 11.5; Table 4: AT-DPT 267.4 vs. DPT 306.8), meaning deploying it when no attacker is present incurs a real cost.
The paper acknowledges this trade-off but does not investigate mitigations.
Separately, the attacker is trained with REINFORCE using a small neural network and has access to the underlying clean reward (Section 3.2, lines 220-222).
The paper does not conduct an ablation study on the attacker's capacity -- it is unknown whether stronger attacker architectures (e.g., PPO, larger networks, or full context access) would break AT-DPT.
If the attacker population is weak, the claimed robustness may be illusory: robust against these specific attackers but not against a truly adversarial worst case.

- **W3. Narrow scope of ICRL backbone and missing experimental comparisons.**
The paper studies robustness exclusively through DPT (Lee et al., 2023), but the ICRL landscape has grown substantially.
Testing whether AT-DPT's adversarial training transfers to at least one alternative backbone (e.g., Algorithm Distillation) is necessary to support the paper's framing as a general solution to ICRL robustness rather than a DPT-specific patch.
Additionally, ARDT (Tang et al., NeurIPS 2024) -- the most directly relevant prior work on adversarial training for decision transformers -- is cited (Section 2, line 124) but never compared experimentally.
The paper states ARDT addresses a "Markov game framework" with "transition probability" modifications, but the core idea of adversarial training for decision transformers is very similar, and a direct comparison (or precise delineation of inapplicability) is needed.

---

> ### Author Rebuttal · Authors · 2026-03-30
>
> We greatly appreciate your time spent reviewing our work and providing valuable feedback. We are happy to see that you found the method well-motivated, thorough and experimentally validated.
>
> W1 & Limitations. We provide an additional set of results in a high-dimensional environment Miniworld (continuous state space, 3D observation), please see Appendix B.4, Table 9. These results show that, similarly to the smaller environments, AT-DPT is more robust to adversarial reward corruption. That is, there is evidence to show that the method scales to larger environments.
>
> W2. We believe that the attacker having access to underlying rewards makes it stronger, and we do not see it as a weakness, but rather a strength. Regarding stronger attackers, we do indeed show experiments with a different attacker architecture: one where the attacker has access to the full context (adaptive attacker, Table 2, lines 330-345). In addition, in Appendix Table 5 we train an AT-DPT agent against a uniform random attack (AT-DPT (rand.) in the table) – comparing it to other robust baselines (e.g., RTS, crUCB) we can see it has a significantly lower regret, and shows evidence for transferring robustness to stronger attacks. Regarding clean environment degradation, algorithms designed specifically for clean settings are expected to perform better there, as they rely on the assumption that there is no adversarial noise. They degrade more severely under adversarial noise: the excess regret they incur relative to our approach in the corrupted setting is significantly larger than the excess regret our method incurs relative to them in the clean setting.
>
> W3. We would like to clarify a misunderstanding of AD [1]. AD is learning to imitate a learning algorithm, meaning it has to have access to trajectories showing successful policy improvement across episodes. In our setting this means that we have to have a good corruption-robust baseline, which is not trivial. We rely on DPT because it doesn’t just distill an existing algorithm, but is trained to predict the optimal action from corrupted experience. Please see an additional discussion on why comparing to AD would not be fair in Appendix A, lines 716-723.
>
> Similarly, ARDT [2] is a method which differs from ours, and we present a precise delineation of inapplicability as follows. They study zero-sum Markov games instead of MDPs; they operate on transition probabilities of the environment, keeping rewards unchanged, instead of studying reward corruption; they train a fixed offline policy to output actions conditioned on a specific target minimax return-to-go token; the agent in ARDT observes the action the adversary took, whereas AT-DPT does not know if poisoning has occurred or not; their setting focuses on single-task performance against an opponent, whereas we focus on the robust multi-task/meta-learning setting with an ‘invisible’ attacker. We hope this clarifies the differences between the methods.
>
> While the landscape of ICRL methods has grown, to our knowledge we are the first to study corruption robustness focusing on the _test-time_ setting, whereas prior work on reward poisoning attacks primarily focuses on training-time attacks. In addition, we are not aware of works prior to ours that considered poisoning attacks for in-context RL. Hence, our framework is novel relative to this line of work as well. While we agree that our method extends DPT by considering adversarial training, we don't fully understand why this would be a weakness of the work. In fact, we would like to argue that this is a principled way of designing robust DPT.
>
> Q1. We provide experiments showing an attacker which has access only to the current state (non-adaptive), and one which has access to the full context (adaptive) in Table 2 (lines 330-345). As noted in our response to W2, in Appendix Table 5 we present results training AT-DPT on a weaker attacker (AT-DPT (rand.) in the table) which yields significantly better performance relative to the baselines when evaluated on a stronger attacker. We also show new experiments how AT-DPT transfers to stronger attacks; please see response to reviewer 1E2G.
>
> Q2. While it is possible to run experiments with corruption level $\varepsilon > 0.5$, it is fundamentally invalid, because a corruption level above 50% means the majority of the data becomes corrupted. If an adversary controls more than 50% of the reward signals they can effectively completely change the apparent distribution, making it mathematically indistinguishable from the true environment [3].
>
> Q3. See response to W3.
>
> We kindly ask the reviewer to consider this response. Thank you again for your feedback and please let us know if you have any additional comments or questions.
>
> [1] Laskin, M. et al. (2023) In-context Reinforcement Learning with Algorithm Distillation.
>
> [2] Tang, X. et al. (2024) Adversarially robust decision transformer.
>
> [3] Donoho, D. L. et al. (1983). The notion of breakdown point.

---

> > ### Author Rebuttal · Reviewer_bk5T · 2026-04-04
> >
> > Thank you for the detailed rebuttal. The additional Miniworld experiments (Table 9) provide helpful evidence that AT-DPT extends to continuous, higher-dimensional environments. The clarifications on cross-attacker generalization and suboptimal-expert robustness are also appreciated.
> >
> > My concerns are adequately addressed. I recommend incorporating the Miniworld results and the suboptimal-expert ablation into the main text for the revision, as they strengthen the paper's scalability and practicality arguments.
> >
> > I will maintain my positive recommendation.

---

> > > ### Author Response · Authors · 2026-04-05
> > >
> > > Thank you for the acknowledgement. We provide a further response to W3.
> > >
> > > We provide an additional set of experiments comparing AD as a baseline to DPT. We train AD with embedding dim. 32, 4 layers, 4 attention heads, 0 dropout, 1e-4 learning rate, AdamW (with 1e-4 weight decay), batch size 64, but the experiment setup is otherwise the same as in Table 1 ($\varepsilon = 0.4$):
> > >
> > > ||AT-DPT|DPT|TS|RTS|UCB|crUCB|Unif. Rand.|Clean|
> > > |-|-|-|-|-|-|-|-|-|
> > > |AD (source TS)|94.8 ± 2.6|84.5 ± 4.7|82.5 ± 4.8|75.0 ± 5.3|78.7 ± 4.4|81.6 ± 1.8|35.5 ± 2.4|9.1 ± 0.6|
> > > |AD (source crUCB)|55.2 ± 2.5|56.1 ± 1.5|57.0 ± 2.2|53.9 ± 1.3|55.3 ± 2.1|52.7 ± 2.0|33.1 ± 1.6|16.1 ± 0.5|
> > > |AD (source crUCB under corruption)|83.9 ± 4.6|80.1 ± 4.4|77.6 ± 3.8|73.9 ± 3.2|76.1 ± 3.4|76.0 ± 3.2|32.9 ± 1.8|15.6 ± 0.5|
> > > |AT-DPT|24.2 ± 1.2|24.8 ± 1.4|29.8 ± 3.0|28.3 ± 1.9|24.5 ± 0.8|23.8 ± 1.4|38.7 ± 1.7|13.0 ± 0.9|
> > > |DPT|63.6 ± 8.6|59.4 ± 5.2|62.0 ± 8.6|59.1 ± 7.3|55.4 ± 8.1|58.8 ± 7.8|37.2 ± 1.2|11.5 ± 0.5|
> > > |TS|106.3 ± 3.8|97.7 ± 4.9|94.3 ± 3.8|93.1 ± 6.0|89.6 ± 1.8|92.6 ± 4.8|34.2 ± 1.6|8.7 ± 0.6|
> > > |crUCB|86.0 ± 4.4|85.0 ± 2.3|82.0 ± 4.4|82.4 ± 3.3|79.4 ± 3.0|82.5 ± 5.1|31.8 ± 1.6|15.8 ± 0.5|
> > >
> > > It seems that AD with a sufficiently good source algorithm (e.g., AD with source crUCB) does perform better than the source algorithm alone (e.g., crUCB), which performs slightly better than simply DPT alone. Out of all of these AT-DPT still seems to have the lowest regret, which is expected, because it is trained to be robust to corruption.
> > >
> > > As mentioned previously, to train robustly for AD we would have to show policy improvement under corruption, which is difficult on its own; alternatively, supplying optimal actions to AD during adversarial training would just make it a DPT method.
> > >
> > > We hope these results strengthen our response to W3, and the reviewer will take them into consideration.

---

### Decision · Program_Chairs · 2026-04-30

**Decision:**

Accept (regular)

**Comment:**

This paper studies test-time reward poisoning in in-context reinforcement learning, focusing on DPT. The reviewers generally agreed that the problem formulation is novel and important: unlike in standard RL, rewards in ICRL are part of the context that directly drives adaptation, so poisoning rewards at test time creates a distinct and meaningful vulnerability. The empirical results also show a consistent positive signal, with AT-DPT outperforming vanilla DPT and several robust baselines across bandits, linear bandits, and simple MDP settings, together with evidence of cross-attacker generalization.

The main strengths are therefore the timeliness of the problem and the usefulness of the empirical contribution. Reviewers found the paper readable and the attacker-defender training setup well motivated. The rebuttal further improved confidence by adding Miniworld results beyond the smallest toy environments, presenting transfer results to stronger attacks, and showing that the method remains reasonably effective under somewhat suboptimal demonstrations. Several reviewers explicitly stated that their concerns were addressed or sufficiently reduced after rebuttal, and the overall final review balance is positive.

At the same time, the paper remains somewhat borderline. The main weaknesses are consistent across the reviews. Methodologically, AT-DPT is best viewed as a natural adversarial-training extension of DPT rather than a fundamentally new algorithmic idea. The experimental evidence, while encouraging, remains largely DPT-centric and still limited in scale relative to the broader motivation. In addition, the framework relies on oracle optimal actions during training, and the paper does not provide formal guarantees for the adversarial training procedure. One reviewer continued to view these scope and breadth issues as significant even after rebuttal.